# ProstaTD: Bridging Surgical Triplet from Classification to Fully Supervised Detection

**Yiliang Chen**[1], **Zhixi Li**[1,2], **Cheng Xu**[1], **Alex Qinyang Liu**[3], **Ruize Cui**[1]
**Xuemiao Xu**[4], **Jeremy Yuen-Chun Teoh**[3], **Shengfeng He**[5,*], **Jing Qin**[1,*]

[1]School of Nursing, The Hong Kong Polytechnic University
[2]Nanfang Hospital, Southern Medical University
[3]Department of Surgery, The Chinese University of Hong Kong
[4]South China University of Technology    [5]Singapore Management University

## Abstract

Surgical triplet detection is a critical task in surgical video analysis, with significant implications for performance assessment and training novice surgeons. However, existing datasets like CholecT50 lack precise spatial bounding box annotations, rendering triplet classification at the image level insufficient for practical applications. The inclusion of bounding box annotations is essential to make this task meaningful, as they provide the spatial context necessary for accurate analysis and improved model generalizability. To address these shortcomings, we introduce *ProstaTD*, a large-scale, multi-institutional dataset for surgical triplet detection, developed from the technically demanding domain of robot-assisted prostatectomy. ProstaTD offers clinically defined temporal boundaries and high-precision bounding box annotations for each structured triplet activity. The dataset comprises 71,775 video frames, collected from 21 surgeries performed across multiple institutions, reflecting a broad range of surgical practices and intraoperative conditions. The annotation process was conducted under rigorous medical supervision and involved more than 60 contributors, including practicing surgeons and medically trained annotators, through multiple iterative phases of labeling and verification. To further facilitate future general-purpose surgical annotation, we developed two tailored labeling tools to improve efficiency and scalability in our annotation workflows. In addition, we created a surgical triplet detection evaluation toolkit that enables standardized and reproducible performance assessment across studies. ProstaTD is the largest and most diverse surgical triplet dataset to date, moving the field from simple classification to full detection with precise spatial and temporal boundaries and thereby providing a robust foundation for fair benchmarking. Our implementation can be found at https://github.com/chen-yiliang/ProstaTD.

## 1 Introduction

Surgical triplet detection is a fundamental task in surgical data science. It involves identifying <instrument, verb, target> triplets, which represent the instrument in use, the action performed, and the anatomical target being acted upon, from each frame of surgical videos. This task supports the development of context-aware decision support systems and contributes to improved surgical safety, procedural standardization, and operational efficiency (Murali et al., 2023; Padoy et al., 2012).

The field was initiated with the release of CholecT40 (Nwoye et al., 2020), the first benchmark for surgical triplet classification in laparoscopic cholecystectomy. It provided frame-level triplet class labels for training models and was later extended through CholecT45 (Nwoye et al., 2023a) and CholecT50 (Nwoye et al., 2022), which included 5 and 10 additional surgical videos, respectively. Currently, CholecT50 is the most widely adopted dataset for the surgical triplet analysis (Xi et al., 2022; Chen et al., 2023; Gui & Wang, 2024). While the CholecTriplet 2022 Challenge (Nwoye et al., 2023b) advanced the field by framing surgical triplet detection as a task involving both recognition

---

*Corresponding authors: shengfenghe@smu.edu.sg and harry.qin@polyu.edu.hk.

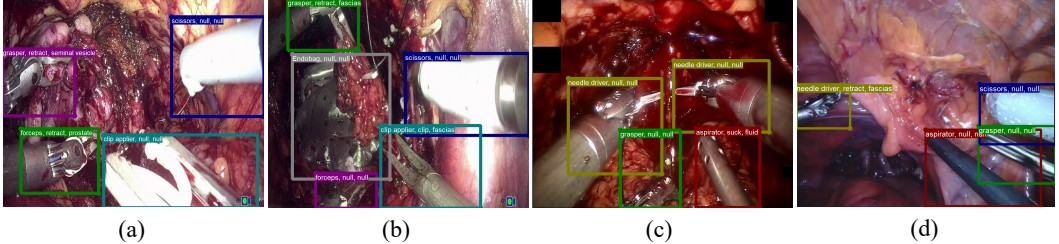

Figure 1: Examples of labeled surgical triplets in our ProstaTD dataset. ProstaTD offers clinically defined temporal boundaries and high-precision bounding box annotations for each structured triplet activity, curated from diverse, multi-institutional surgical sources, establishing a strong foundation for precise and robust triplet detection. Here, (a) and (b) are examples from our in-house collected videos called Prostate Wellness Hope (PWH) dataset, (c) from the PSI-AVA dataset, and (d) from the ESAD dataset. *All videos were comprehensively annotated by our team.*

and spatial localization, it relied primarily on weak supervision with frame-level triplet class labels from the CholecT50 dataset. This constraint limited the ability to achieve precise spatial localization, often resulting in ambiguous triplet predictions.

Despite these contributions, CholecT50 (Nwoye et al., 2022) has three critical limitations:

1. **No bounding box annotations:** CholecT50 provides only class labels without spatial localization, which restricts the task to weakly supervised settings and limits localization accuracy.

2. **Unclear and inconsistent temporal boundaries:** Although the CholecT50 paper briefly states that its annotators marked the start and end of each triplet, the procedure lacks detailed specification. It remains ambiguous whether a new triplet activity begins with instrument entry or target contact, and whether it ends with instrument disengagement, a change in the target being operated on, or instrument exit. Such uncertainty leads to inconsistent labeling and limits the model's ability to capture temporal dynamics and instrument–target interactions.

3. **Lack of diversity in data sources:** CholecT50 is collected at a single institution, limiting variation in workflow and instrument appearance. Differences in manufacturers, as well as surgeons' habits and educational backgrounds, produce rare triplets that are often missing, causing models trained on this dataset to overfit to local style and generalize poorly to other clinical settings.

To address these limitations, we develop the first open-source annotation tools specifically designed for surgical triplet annotation. Building upon this, we introduce *ProstaTD*, the first large-scale surgical triplet detection dataset encompassing full-length surgical procedures with precise multi-instrument localization. ProstaTD enables simultaneous and reliable recognition and localization of surgical triplets, a capability essential for modeling instrument-tissue interactions and developing intelligent systems for intraoperative guidance. The dataset includes 21 robot-assisted prostatectomy videos, totaling 71,775 annotated frames. These were sourced from three heterogeneous domains: 4 videos from the publicly available ESAD (Bawa et al., 2021), 8 from the PSI-AVA (Valderrama et al., 2022) dataset, and 9 newly acquired from our in-house dataset termed Prostate Wellness Hope (PWH) dataset. Compared to existing cholecystectomy datasets, ProstaTD offers higher instrument concurrency, greater anatomical complexity, and increased domain diversity. These characteristics better reflect the variability and challenges of real-world high-complexity surgical environments. Each frame in ProstaTD is annotated with structured triplet information, including precise bounding boxes for all visible instruments, detailed action (verb) labels, and anatomical target identifiers. All annotations are constrained within expert-defined temporal boundaries curated by experienced urologists, ensuring both clinical accuracy and temporal consistency. Fig. 1 showcases representative examples of the annotated triplets in ProstaTD.

To enable comprehensive and efficient comparisons, we develop an evaluation toolkit and conduct extensive benchmarks using state-of-the-art models. We also propose a tailored baseline method that applies a distillation mechanism at the instance level to mitigate triplet imbalance and facilitate fair comparison in future work. Results show that the fine-grained spatial annotations in ProstaTD significantly improve the performance and robustness of models for surgical triplet detection. ProstaTD thus provides a comprehensive foundation for advancing surgical video analysis.

Table 1: Comparison of existing surgical datasets with ProstaTD, highlighting the unique attributes required for surgical triplet detection. LC and RARP denote Laparoscopic Cholecystectomy and Robot-Assisted Radical Prostatectomy. Full BBox indicates that every frame is fully annotated with bounding boxes. Complete Surgery denotes that the dataset as a whole provides coverage for the entire surgical procedure. Triplet Boundary refers to the temporal start and end range of a complete surgical triplet. Multiple Sources indicates that the data is collected from various hospital.

| Dataset | Task | Attributes | | | | | | Statistics | |
|---|---|---|---|---|---|---|---|---|---|
| | | Supervised Detection | Full BBox | Triplet-like Structure | Triplet Boundary | Multiple Sources | Complete Surgery | No. Instances | No. Triplets |
| Cholec80-locations | LC | ✓ | ✓ | | | | ✓ | 6,471 | – |
| CholecTrack20 | LC | ✓ | ✓ | | | | ✓ | 65,200 | – |
| ESAD | RARP | ✓ | ✓ | | | | ✓ | 46,753 | – |
| PSI-AVA | RARP | ✓ | | | | | ✓ | 5,804 | – |
| CholecQ | LC | ✓ | ✓ | ✓ | | | | 14,480 | 17 |
| CholecT45 | LC | | | ✓ | | | ✓ | 146,394 | 100 |
| CholecT50 | LC | | | ✓ | | | ✓ | 161,988 | 100 |
| ProstaTD (Ours) | RARP | ✓ | ✓ | ✓ | ✓ | ✓ | ✓ | 196,490 | 89 |

In summary, our contributions are fourfold:

- We introduce a new task with a new dataset for fully supervised surgical triplet detection at the procedure level. To the best of our knowledge, our ProstaTD is the *largest* surgical dataset with instance-level annotations.

- We construct a multi-institutional surgical triplet dataset featuring detailed labels (precise bounding boxes and standardized triplet boundaries) in more complex surgical scenarios.

- We release two open-source annotation tools, which are the first specifically tailored for surgical triplet annotation, along with an open-source evaluation toolkit for benchmarking surgical triplet detection, providing a foundation for surgical triplet analysis across diverse surgical procedures.

- We introduce the first benchmark for fully supervised surgical triplet detection, providing our tailored method as a baseline for comparison.

## 2 RELATED WORK

While many surgical video datasets have emerged, most are not designed to support triplet-based reasoning. Datasets such as Cholec80-locations (Shi et al., 2020) and CholecTrack20 (Nwoye et al., 2025) provide bounding box annotations but focus primarily on instrument tracking, without structured action-target associations required for triplet detection. A few datasets offer partial triplet-style annotations but are limited in scope, accessibility, or domain relevance. For instance, MISAW (Huaulmé et al., 2021) is restricted to simulated anastomosis tasks. RLLS-I2M (Zhao et al., 2022), Cataract (Lin et al., 2022), and PhacoQ (Lin et al., 2024) are not publicly available, making comparative analysis infeasible. CholecQ (Lin et al., 2024) is a notable exception, offering bounding box annotations for triplet-like structures. However, it is constrained to three-second clips from Cholec80 (Twinanda et al., 2016), resulting in high frame redundancy and limited temporal diversity. Consequently, their triplet boundaries can't cover a complete action, so the dataset does not qualify as a resource for proper triplet action modeling. Moreover, this limitation also results in the dataset lacking full procedure coverage and containing only 17 unique triplet types, far fewer than the 100 in CholecT50 or the 89 in our proposed ProstaTD. These limitations render CholecQ a toy-scale dataset that is inadequate for robust real-world triplet modeling (see Table 1).

To our knowledge, CholecT45 and CholecT50 were the only datasets used for surgical triplet research at the procedure level prior to ProstaTD. They originate from a subset of cholecystectomy videos in Cholec80, and comprise 146,394 and 161,988 annotated triplet instances across 100 classes, respectively. However, the publicly available labels are provided only at the frame level and do not include spatial bounding boxes or explicitly defined temporal boundaries for triplets. Although bounding boxes exist for the test set in the CholecTriplet2022 challenge (Nwoye et al., 2023b),

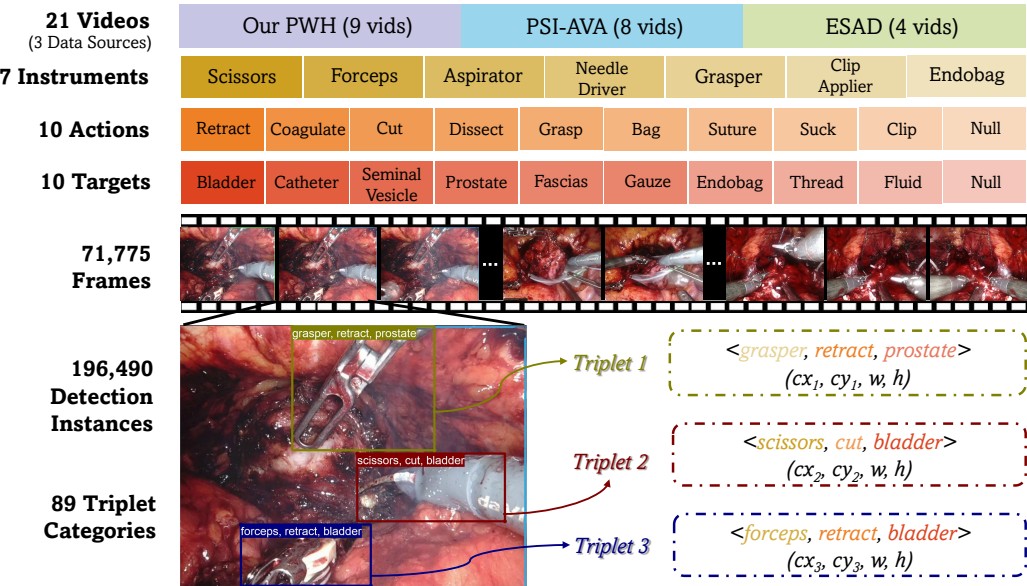

Figure 2: Overview of the ProstaTD dataset structure. The dataset consists of 21 videos curated from three sources: our in-house PWH, PSI-AVA (Valderrama et al., 2022), and ESAD (Bawa et al., 2021). *All videos were comprehensively annotated by our team*, covering 7 instruments, 10 actions, and 10 targets. In total, the dataset contains 71,775 frames, 196,490 annotated instances, and 89 surgical triplet categories, with each frame labeled using precise bounding boxes.

the test annotations are not publicly accessible. Participants were required to submit predictions for server-side evaluation to obtain results, without direct access to the ground truth. Therefore, CholecT45 and CholecT50 support only surgical triplet classification rather than triplet detection.

The two prostatectomy resources, ESAD (Bawa et al., 2021) and PSI-AVA (Valderrama et al., 2022), provide RARP videos, but their original annotations were not designed for surgical triplet detection. ESAD includes full bounding boxes, yet many are coarse or group multiple instruments within a single bounding box, while PSI-AVA contains only sparsely annotated frames rather than dense frame-by-frame labels. As for other public prostatectomy datasets such as GraSP (Ayobi et al., 2025), TAPIS (Ayobi et al., 2023), and SAR-RARP50 (Psychogyios et al., 2023), they share substantial overlap with PSI-AVA videos and were therefore not incorporated into our ProstaTD dataset.

The proposed ProstaTD addresses these limitations and is specifically designed for triplet detection in high-complexity procedures. It is substantially larger than existing datasets, featuring 196,490 annotated surgical triplet instances (see Fig. 2). In addition, it offers dense spatial annotations, clinically standardized temporal triplet boundaries, and data collected from multiple institutions. These attributes enable precise localization, consistent temporal labeling, and broad generalizability, which together support more robust and structured analysis of surgery.

## 3 PROSTATD DATASET

### 3.1 PROBLEM DEFINITION OF SURGICAL TRIPLET DETECTION

Given a video dataset $V = \{V_1, V_2, \ldots, V_n\}$ of laparoscopic surgeries, each video $V_i$ contains a sequence of frames $F = \{F_1, F_2, \ldots, F_m\}$. Each frame $F_t$ may include multiple instrument instances, annotated with a bounding box $B = \{(c_x, c_y, w, h)\}$, where $(c_x, c_y)$ is the center, and $w, h$ denote width and height. Each instance has a triplet class label $C \in \{C_1, C_2, \ldots, C_N\}$, representing a specific instrument-action-target combination, with $N$ as the total number of triplets defined by the task. The goal of surgical triplet detection is to identify all valid triplets $\mathcal{T}_t = \{C_1, C_2, \ldots, C_k\}$ in each frame $F_t$ providing both the spatial location and the semantic label for each interaction.

### 3.2 PROSTATD DATA FORMAT

For benchmarking, we release annotations in both COCO and YOLO formats. In the YOLO version (see Fig. 2), each annotated instance is represented as $\langle tri, i, a, t, c_x, c_y, w, h \rangle$, where $tri, i, a, t$ denote the triplet, instrument, action, and target identifiers, and $(c_x, c_y, w, h)$ denote the bounding box center, width, and height. This representation follows the standard convention commonly adopted in YOLO-series models. In addition, we provide the COCO format, where $(x, y, w, h)$ specifies the top-left corner, width, and height, which is widely used by other detection frameworks. By offering both formats, we ensure compatibility with a broad range of models and evaluation pipelines.

### 3.3 DATA COLLECTION

Our dataset, illustrated in Fig. 2, integrates three sources: our in-house PWH Dataset, the publicly available PSI-AVA dataset and the ESAD dataset. Here, ESAD is released under the CC-BY-NC-SA-4.0 license and PSI-AVA is hosted by MIT. The PWH videos were recorded with institutional ethical approval, and the final release of our dataset will also comply with the CC-BY-NC-SA-4.0 license. Notably, we did not utilize any existing annotations from ESAD or PSI-AVA, even when they were partially relevant to our task. For further details on the original annotations of these datasets, please refer to Appendix A.1. This decision was driven by differences in data sources and the need for a unified annotation protocol to ensure consistent labeling of surgical actions, targets, and bounding boxes. Accordingly, guided by the expert insights of our team's urologist, we developed and strictly followed our own annotation criteria throughout the process.

### 3.4 ANNOTATION DESIGN

**Categories Design.** As shown in Fig. 2, our annotation schema comprises 7 surgical instruments, 10 actions, and 10 targets. These categories were determined through consensus among ten professional surgeons, with some labels merged to ensure consistency and reduce redundancy. For details on how labels were merged, please refer to Appendix A.4. For surgical instrument bounding boxes, we adopted a consistent strategy: for instruments with well-defined contours, only the head and joint regions were enclosed, whereas instruments with indistinct boundaries, such as "aspirator", were annotated using full-enclosure bounding boxes. These annotation rules were consistently applied across all instrument categories, ensuring uniform quality and reducing ambiguity in model training.

**Temporal Boundaries of Triplets.** Unlike CholecT50 (Nwoye et al., 2022), we established unified guidelines to define the temporal boundaries of surgical triplets. Importantly, triplet starting points were not annotated based solely on the presence of a surgical instrument in the frame, as such an approach resembles instance-level step recognition and lacks alignment with clinical assessment of surgical skill. Following collaborative discussions, we categorized triplets into three types: continuous actions, momentary actions, and null actions. For continuous actions, such as lymph node dissection along the iliac vessels, the action was defined to start only when the surgical instrument made contact with or was extremely close to the target, and to end when the instrument departed from the same target for more than 2 seconds. For momentary actions, such as cutting sutures with scissors, the temporal window was defined more flexibly: annotations included up to 2 seconds before and after the moment of instrument–target contact. The remaining cases were designated as null actions, where the triplet corresponded to an instrument being stationary or moving without meaningful interaction. This clinically informed strategy ensures temporally precise and semantically meaningful annotation of triplet actions, better aligning with real-world surgical practices. Owing to these detailed rules, we were able to consolidate annotations from multiple annotators with consistent quality, which is further supported by the inter-rater agreement results in Section 5.

### 3.5 ANNOTATION PROCESS

**Labeling Tools.** We developed two dedicated annotation applications for ProstaTD: *Triplet-labelme*, designed for single-frame triplet editing *(instrument, action, target)* with bounding box assignment, and *SurgLabel*, tailored for high-throughput batch labeling of actions and targets across user-defined temporal ranges. Both tools will be released as open source to support large-scale triplet annotation across diverse surgical procedures, and they are also broadly applicable to general surgical video annotation, contributing to foundation research in this domain. Please see Appendix C for details.

Table 2: Percentage distribution of triplet instances per frame in surgical triplet datasets between CholecT50 (Nwoye et al., 2022) and our ProstaTD.

| Dataset | 0 | 1 | 2 | 3 | 4 | 5 | 6 |
|---|---|---|---|---|---|---|---|
| CholecT50 (Nwoye et al., 2022) | 10.97% | 35.18% | 47.10% | 6.75% | 0.00% | 0.00% | 0.00% |
| ProstaTD (Ours) | 0.91% | 6.19% | 34.13% | 39.75% | 16.62% | 2.21% | 0.19% |

**Semi-Automatic Annotation for Surgical Instruments.** In the initial phase of annotation, we focused on labeling the types and bounding boxes of surgical instruments used in prostatectomy procedures. To support this task, we recruited 43 additional part-time student assistants, resulting in a total of 48 annotators. Among them, 20 students with strong medical backgrounds were assigned to conduct multiple rounds of review and quality control. Initial annotations were generated based on pre-labeled results from our previously developed cystoscopic surgical instrument detection model. Although bladder and prostate surgeries share visual similarities, domain-specific differences require extensive manual refinement. Consequently, our team performed at least three rounds of manual corrections on selected videos before using them to fine-tune the instrument detection model for subsequent annotation generation. After five iterative cycles of this semi-automatic training and correction process, we successfully completed both instrument type classification and bounding box annotations for all surgical instruments across the entire video dataset.

**Surgical Action and Target Labeling.** In the second phase, we enriched the instrument annotations from the first phase by assigning corresponding actions and targets. Unlike the first phase, which could be carried out by student annotators, this stage required substantial medical expertise. Although we employed the batch annotation tool, the annotation and refinement process were considerably more time-consuming. This phase was conducted by ten surgeons and fourteen senior students with strong medical backgrounds. To ensure label accuracy, each frame was reviewed by at least three team members. This process resulted in a high-quality ProstaTD dataset comprising 89 triplet categories and 196,490 annotated instances, as illustrated in Fig. 2.

## 4 ANALYSIS OF THE PROSTATD DATASET

### 4.1 COMPLEXITY ANALYSIS OF PROSTATD

According to publicly available surgical classification systems (e.g., those used by regional health authorities (Hospital Authority, 2025)), cholecystectomy is typically categorized as a major procedure, whereas prostatectomy is considered ultra-major, reflecting higher surgical complexity. This distinction highlights the increased technical demands and environmental intricacy associated with prostatectomy procedures. The complexity gap is quantitatively reflected in Table 2, which compares the distribution of concurrent triplet instances in CholecT50 (cholecystectomy) and our proposed ProstaTD (prostatectomy) datasets. As summarized in Table 2, CholecT50 has 93.25% of frames with $\leq 2$ triplet instances and none with $\geq 4$, whereas ProstaTD has 58.77% of frames with $\geq 3$ instances. This denser, more crowded scene composition underscores the higher complexity of prostatectomy and poses greater challenges for triplet detection models.

### 4.2 DISTRIBUTION OF PROSTATD DATASET

**Distribution of Individual Components.** As shown in Fig. 3, component distributions in ProstaTD align with the procedure's two main phases: dissection and anastomosis. Instrument frequencies reflect this structure: dissection instruments like monopolar scissors and bipolar forceps are most common, while needle drivers peak during anastomosis. Aspirators and graspers are used consistently across both phases for suction and retraction. Endobags and clip appliers are infrequent, as they are restricted to highly specific steps. The action distribution is dominated by *retract*, reflecting the need to maintain field clarity. The target distribution extends beyond anatomy to include catheters, gauze, and endobags, enhancing clinical realism. Samples may also contain *null* actions or targets, indicating an instrument is stationary or moving without a meaningful surgical action. Additionally, a detailed analysis of instrument variations discussed in Appendix A.3.

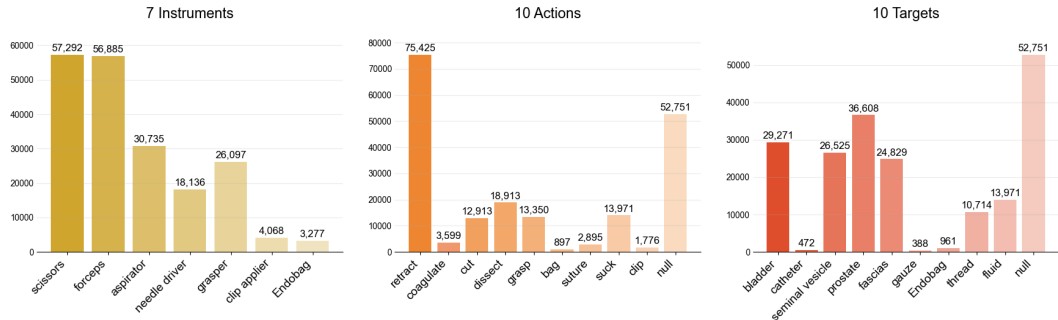

Figure 3: Component statistics in ProstaTD showing instruments (left), actions (center), and targets (right). Counts reflect label frequency and illustrate dataset diversity and task complexity.

Table 3: Cohen's Kappa between independent surgeon reviews and consolidated ground truth on five randomly selected 200-frame segments. The bold value highlights the average result.

|               | Surgeon 1 | Surgeon 2 | Surgeon 3 | Surgeon 4 | Surgeon 5 | **Average** |
|---------------|-----------|-----------|-----------|-----------|-----------|-------------|
| Cohen's Kappa | 0.82      | 0.81      | 0.84      | 0.82      | 0.83      | **0.82**    |

**Distribution of Triplet Components.** As shown in Table 6 in the Appendix, triplet frequencies differ noticeably across ESAD, PSI, and PWH. Integrating these distinct sources is essential for capturing clinical variability, which arises from factors like differing surgical preferences, institutional training protocols, and patient-specific conditions. For instance, ESAD features a higher frequency of fascia-related triplets, while PSI-AVA shows prominent vesicle and aspirator retraction patterns. In PWH, endobag interactions are common, reflecting a technique where surgeons position the bag centrally for easier access. Capturing this diversity across sources is crucial for robust model generalization. Further qualitative insights into these distributional differences are discussed in Appendix B.

## 5 INTER-RATER AGREEMENT

To assess annotation quality, we conducted an inter-rater agreement study. We randomly sampled five non-overlapping video segments of 200 frames each. Five experienced surgeons who were not involved in the original annotation independently reviewed the triplet labels. Each surgeon assessed one segment while viewing the bounding box annotations but blinded to the triplet categories. Agreement with the consolidated ground truth was quantified using Cohen's Kappa, with results summarized in Table 3. The average Kappa was 0.82, indicating strong consistency with our annotations. Most discrepancies occurred near anatomical boundaries where target assignment is inherently ambiguous; these cases are mitigated by our detailed written guidelines and expert-driven review protocol. Furthermore, all disputed cases were resolved through a secondary consensus round among senior surgeons, and the final ground truth was refined accordingly to maximize reliability.

## 6 EXPERIMENTS

### 6.1 EVALUATION PROTOCOL

To rigorously assess our dataset and models, we adopt a five-fold cross-validation over all 21 surgical videos, rotating the test fold and using the remaining videos for training and validation following our setting. In addition, we introduce an evaluation toolkit ("**ivtdmetrics**" in Appendix D) tailored for surgical triplet detection benchmarking, supporting mAP at IoU 0.5 and 0.5–0.95, as well as Precision, Recall, and F1-score, thereby facilitating fair comparison today and providing a basis for future foundation model research. For a detailed five-fold composition, please refer to Appendix B.2.

Table 4: Detection performance on the ProstaTD dataset for I, V, T, and IVT components. We report mAP at IoU thresholds of 50% ("50") and 50:95 ("95"), together with inference speed (FPS). All results are reported as mean$_{\pm\text{std}}$ (%) over 5-fold cross-validation. Experiments are conducted with input size $640\times640$ on a single NVIDIA RTX 4090 GPU. Bold values with light green background indicate the best results, and underlined values with light purple background indicate the second-best.

| Method | $mAP_I(\%)\uparrow$ | | $mAP_V(\%)\uparrow$ | | $mAP_T(\%)\uparrow$ | | $mAP_{IVT}(\%)\uparrow$ | | FPS $\uparrow$ |
|---|---|---|---|---|---|---|---|---|---|
| | 50 | 95 | 50 | 95 | 50 | 95 | 50 | 95 | |
| Tripnet-Det[*] | $1.6_{0.4}$ | – | $0.6_{0.3}$ | – | $0.4_{0.1}$ | – | $0.1_{0.0}$ | – | **331.8** |
| RDV-Det[*] | $1.8_{0.5}$ | – | $0.6_{0.4}$ | – | $0.3_{0.1}$ | – | $0.1_{0.0}$ | – | 146.6 |
| Faster R-CNN | $73.3_{4.9}$ | $63.2_{5.4}$ | $48.4_{5.8}$ | $42.1_{5.3}$ | $43.5_{6.3}$ | $37.6_{5.4}$ | $25.9_{4.4}$ | $22.6_{3.9}$ | 23.4 |
| Cascade R-CNN | $69.5_{5.1}$ | $59.6_{5.6}$ | $44.6_{6.1}$ | $38.5_{5.6}$ | $39.5_{6.6}$ | $33.6_{5.6}$ | $21.6_{4.6}$ | $18.7_{4.1}$ | 20.6 |
| SSD | $74.6_{4.8}$ | $64.5_{5.3}$ | $50.2_{5.7}$ | $43.7_{5.2}$ | $45.4_{6.1}$ | $39.4_{5.2}$ | $27.1_{4.3}$ | $23.8_{3.9}$ | 82.4 |
| Vit-Det | $86.5_{2.8}$ | $73.6_{2.8}$ | $52.2_{4.6}$ | $45.4_{3.8}$ | $48.1_{4.8}$ | $41.2_{3.8}$ | $30.2_{3.9}$ | $26.8_{3.5}$ | 16.8 |
| Deformable-DETR | $75.4_{4.7}$ | $65.0_{5.2}$ | $51.1_{5.7}$ | $44.5_{5.1}$ | $46.3_{6.2}$ | $40.1_{5.2}$ | $27.5_{4.6}$ | $24.0_{4.1}$ | 24.5 |
| RT-DETR | $\mathbf{91.6_{0.9}}$ | $\mathbf{81.0_{1.6}}$ | $58.9_{4.7}$ | $52.8_{3.3}$ | $\mathbf{56.8_{2.4}}$ | $50.6_{2.2}$ | $33.0_{3.8}$ | $29.6_{3.3}$ | 66.3 |
| YOLOv10 | $88.4_{1.3}$ | $80.7_{2.2}$ | $59.4_{3.4}$ | $\underline{54.9_{2.5}}$ | $54.6_{3.2}$ | $50.2_{3.1}$ | $34.3_{4.1}$ | $\underline{31.8_{3.5}}$ | 200.3 |
| YOLOv11 | $88.2_{1.4}$ | $80.0_{2.5}$ | $59.1_{3.2}$ | $54.4_{2.1}$ | $55.6_{3.6}$ | $\mathbf{51.0_{3.5}}$ | $34.1_{3.7}$ | $31.5_{3.3}$ | 185.2 |
| YOLOv12 | $88.8_{1.1}$ | $80.4_{1.9}$ | $\underline{59.9_{3.1}}$ | $54.8_{2.2}$ | $54.5_{2.1}$ | $49.9_{1.9}$ | $\underline{34.3_{3.8}}$ | $31.5_{3.2}$ | $\underline{204.1}$ |
| TAPIR | $76.1_{4.5}$ | $65.8_{4.8}$ | $52.3_{5.1}$ | $45.6_{4.6}$ | $47.1_{5.4}$ | $40.5_{4.7}$ | $28.4_{4.7}$ | $24.6_{4.2}$ | 10.6 |
| MCIT-IG | $77.4_{4.4}$ | $67.2_{4.6}$ | $53.6_{4.9}$ | $46.9_{4.3}$ | $48.5_{5.1}$ | $41.8_{4.5}$ | $29.6_{4.5}$ | $26.0_{4.0}$ | 16.0 |
| TDnet (Ours) | $\underline{89.9_{1.3}}$ | $\underline{81.0_{2.0}}$ | $\mathbf{61.7_{2.9}}$ | $\mathbf{56.3_{2.1}}$ | $\underline{55.7_{2.4}}$ | $\underline{50.8_{2.7}}$ | $\mathbf{36.1_{3.4}}$ | $\mathbf{33.1_{3.1}}$ | 126.6 |

[*] Weakly-supervised methods.

## 6.2 COMPARISON METHODS

We establish a benchmark on the proposed ProstaTD dataset by training representative methods, as summarized in Table 4, with implementation details provided in Appendix F.1. Note that prior triplet classification approaches are not directly comparable, as they do not output the bounding boxes required for triplet detection. To examine the gap between weakly-supervised and fully-supervised settings, we include two Cholectriplet2022 challenge methods, Tripnet-Det and RDV-Det (Nwoye et al., 2023b). Both rely solely on class labels for weak supervision, employing Class Activation Maps and Non-Maximum Suppression for triplet localization. Other challenge entries are excluded due to incomplete methodological descriptions, unavailable code, and inferior reported performance. These two weakly-supervised baselines are primarily included to highlight the performance gap relative to fully-supervised detectors with bounding box supervision.

We further compare ProstaTD performance using advanced detection architectures. We first include several classical baselines such as Faster R-CNN (Ren et al., 2016), Cascade R-CNN (Cai & Vasconcelos, 2018), SSD (Liu et al., 2016), and ViT-Det (Li et al., 2022). We then evaluate the state-of-the-art detectors, including Deformable-DETR (Zhu et al., 2021) and RT-DETR (Zhao et al., 2024), as well as the latest YOLO architectures YOLOv10 (Wang et al., 2024), YOLOv11 (Khanam & Hussain, 2024), and YOLOv12 (Tian et al., 2025). In addition, we include TAPIR (Valderrama et al., 2022), a framework that leverages temporal information for multi-task surgical scene detection, where we extend it with an additional triplet branch and remove unrelated tasks. We also evaluate MCIT-IG (Sharma et al., 2023), originally designed for semi-supervised surgical triplet detection, which we train in a fully-supervised manner on ProstaTD in Table 4. Finally, we present a tailored method named TDnet to provide insights for future research. TDnet adopts a multi-task learning strategy and integrates self-distillation in triplet detection. Further details are provided in Appendix E.

## 6.3 RESULTS AND DISCUSSION

**Detection Performance Analysis.** Among all metrics, mAP$_{IVT}$ is the most critical as it evaluates complete triplets. As summarized in Table 4, the two weakly-supervised pipelines, Tripnet-Det and RDV-Det, yield extremely low mAP scores, consistent with the results and findings in the CholecTriplet2022 challenge (Nwoye et al., 2023b). This poor performance is mainly due to their reliance on class labels that provide only weak supervision, which cannot disentangle co-occurring instruments or reliably associate instruments with their corresponding actions and targets.

Table 5: Performance of the IVT component on the ProstaTD test set, reporting Precision, Recall, and $F_1$. Results are mean$_{\pm\text{std}}$ (%) over 5-fold cross-validation. Bold values with light green background indicate the best results, and underlined values with light purple background indicate the second-best.

| Method | Precision ↑ | Recall ↑ | F1-score ↑ |
|---|---|---|---|
| Deformable DETR (Zhu et al., 2021) | $36.1_{4.8}$ | $19.7_{2.8}$ | $22.7_{3.8}$ |
| RT-DETR (Zhao et al., 2024) | $\mathbf{36.4}_{2.9}$ | $31.5_{4.2}$ | $30.9_{3.7}$ |
| YOLOv10 (Wang et al., 2024) | $34.6_{3.8}$ | $34.8_{3.6}$ | $31.9_{3.2}$ |
| YOLOv11 (Khanam & Hussain, 2024) | $33.5_{3.4}$ | $36.1_{3.8}$ | $31.5_{3.3}$ |
| YOLOv12 (Tian et al., 2025) | $33.5_{4.6}$ | $36.2_{4.0}$ | $31.9_{4.0}$ |
| TAPIR (Valderrama et al., 2022) | $35.2_{5.0}$ | $20.3_{3.8}$ | $23.4_{4.3}$ |
| MCIT-IG (Sharma et al., 2023) | $35.5_{4.8}$ | $21.0_{3.6}$ | $24.1_{4.2}$ |
| TDnet (Ours) | $34.7_{4.3}$ | $\mathbf{39.7}_{4.3}$ | $\mathbf{32.8}_{3.6}$ |

Conventional detectors such as SSD, Faster R-CNN, Cascade R-CNN, ViT-Det, and Deformable-DETR demonstrate basic recognition ability, but still show a large gap compared with modern SOTA models. Among advanced detectors, including YOLOv10, YOLOv11, YOLOv12, and RT-DETR, results are generally comparable. Notably, RT-DETR shows strong potential by achieving the highest instrument score (mAP$_I$@0.5 = 91.6%), indicating that its transformer-based design can effectively capture instrument-level features. However, this advantage comes at the cost of slower inference speed (66.3 FPS), which limits its practical deployment in real-time surgical applications. Regarding models tailored to surgical analysis, TAPIR underperforms despite its use of temporal information. This is largely due to the reliance on sparse annotations for temporal features and the outdated detector backbone. Similarly, MCIT-IG, originally designed as a semi-supervised two-stage framework for action triplet detection, does not perform effectively under fully-supervised training on our dataset, likely because its architecture is optimized for semi-supervised settings and specialized modules hinder efficiency when labels are complete.

Our proposed TDnet achieves the best results across all major components. It reaches 36.1% mAP$_{IVT}$@0.5 and 33.1% mAP$_{IVT}$@0.50:0.95, improving upon YOLOv12 from 34.3% to 36.1% and from 31.8% to 33.1%, respectively. In terms of individual components, TDnet attains the highest mAP$_V$ and delivers competitive gains on both instrument and target. It also maintains a strong balance between accuracy and efficiency, achieving 126.6 FPS. Notably, as discussed in Appendix F.2, its advantage becomes even more pronounced under the video-wise mAP evaluation protocol, further highlighting the robustness of our approach across both frame-wise and video-wise assessments.

**Precision–Recall Analysis.** As shown in Table 5, class imbalance in the dataset makes surgical triplet detection highly challenging, leading to generally low $F_1$-scores across all methods. Deformable DETR lags behind more advanced approaches, with an $F_1$ of only 22.7%, highlighting its difficulty in handling long-tail distributions. YOLOv10, YOLOv11, and YOLOv12 achieve balanced precision and recall in the mid-30% range, yielding $F_1$ around 32%. RT-DETR shows the highest precision (36.4%) but relatively low recall (31.5%), which limits its $F_1$ to 30.9%. TAPIR and MCIT-IG also obtain competitive precision (35.2% and 35.5%) but their recall remains below 22%, resulting in very low $F_1$ scores (23.4% and 24.1%). In contrast, our proposed TDnet achieves the best balance, boosting recall to 39.7% while maintaining a precision of 34.7%. This trade-off results in the highest $F_1$ (32.8%), surpassing all baselines. The gain demonstrates that TDnet mitigates the imbalance problem by capturing more true positives without an excessive rise in false positives. Despite these improvements, the absolute scores highlight that surgical triplet detection under heavy class imbalance remains far from solved, leaving considerable space for future progress.

**Additional Analysis.** Beyond the main comparison, we conducted further evaluations, including video-wise metric analysis, ablation studies, confusion matrix analysis, and per-class AP analysis, to demonstrate the effectiveness of our method in mitigating the challenges of triplet prediction. The detailed results of these additional experiments are provided in Appendix F.

## 7 CONCLUSION

Inspired by the pioneering and influential CholecT50 work from the CAMMA team, We introduced *ProstaTD*, the first fully-supervised dataset for surgical triplet detection at the procedure level. It

contains 71,775 frames from 21 multi-institutional surgeries, each with bounding box supervision and clinically verified temporal boundaries. ProstaTD is accompanied by open-source annotation tools, a standardized evaluation toolkit, and baseline benchmarks, forming a comprehensive resource for fair and reproducible research in surgical video analysis. To provide a strong reference for future work, we proposed TDnet, a baseline method that applies instance-level self-distillation with auxiliary supervision on instruments, actions, and targets. This design alleviates class imbalance and improves the robustness of triplet detection, offering a practical starting point for model development on ProstaTD. Overall, ProstaTD moves the field from coarse triplet classification toward full detection with spatial precision.

## 8 REPRODUCIBILITY STATEMENT

Our implementation can be found at https://github.com/chen-yiliang/ProstaTD.

## 9 ETHICAL STATEMENT

All procedures performed in this study with human participants were in accordance with ethical standards and approved by the Institutional Review Board (IRB). The CREC ethics numebr is CREC Ref. No.:2021.650-T. All collected data was thoroughly anonymized and stored in secure facilities with restricted access limited to authorized research team members. The study adhered to all relevant data protection regulations and medical research guidelines throughout data collection, processing, and analysis phases. In the following, we discuss the potential positive and negative societal impacts of our work.

**Potential Benefits.** As a comprehensive dataset for precise bounding box annotations and multi-institutional data, ProstaTD has the potential to significantly advance surgical AI development. The fine-grained annotations enable AI systems to monitor instrument-tissue interactions in real-time, generating objective skill assessments for residents and providing data-driven feedback for continuous surgical quality improvement. Our open-source, multi-center dataset significantly reduces single-source bias and promotes model generalization across different hospitals and surgical settings, establishing a new standard for surgical AI validation. This dataset empowers surgical robot manufacturers to pre-train their vision modules directly, potentially accelerating clinical translation and reducing development cycles by months or even years. Moreover, the comprehensive nature of ProstaTD could catalyze breakthrough innovations in surgical automation and safety systems, ultimately contributing to more standardized and safer surgical procedures worldwide.

**Potential Risks.** The dataset might be exploited by unregulated third parties for commercial products, where model failures could directly compromise patient safety. *Mitigation Measures:* (1) We implement CC-BY-NC-SA-4.0 license requiring all derivatives to maintain non-commercial terms; (2) We recommend incorporating uncertainty estimation and human oversight prompts when deploying models. (3) We explicitly prohibit unauthorized use of the dataset for training LLMs or commercial AI systems, and reserve the right to take legal action against any violations of these terms.

ACKNOWLEDGMENTS

We sincerely thank all the annotators from The Hong Kong Polytechnic University (PolyU), Nanfang Hospital Southern Medical University, Prince of Wales Hospital (PWH), South China University of Technology (SCUT) and The University of Hong Kong (HKU) for their dedicated efforts and valuable contributions. This work was supported by a Shenzhen-Hong Kong-Macao Science and Technology Plan Project (Category C Project) under Shenzhen Municipal Science and Technology Innovation Commission (project no SGDX20230821092359002), a grant under Collaborative Research Fund of Hong Kong Research Grants Council (project no C5055-24G), the Strategic Topics Grant 2025/26, Research Grants Council, HKSAR (Reference no.: STG1/M-405/25-N), the Guangdong Natural Science Funds for Distinguished Young Scholars (No. 2023B1515020097), and the Lee Kong Chian Fellowships. This work was also supported by the Key-Area Research and Development Program of Guangzhou City (No. 2023B01J0022) and the NSFC Key Project (No. U23A20391).

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

# CONTENTS OF APPENDIX

## A    PROSTATD DATASET PREPARATION

In this section, we first provide an additional introduction to the ESAD and PSI-AVA datasets, then we describe the video preprocessing pipeline to ensure data quality, analyze surgical instrument variations across sources, and conclude with the expert-guided strategy for label consolidation.

### A.1    ADDITIONAL INTRODUCTION TO ESAD AND PSI-AVA DATASETS

Both the ESAD and PSI-AVA datasets are collected from prostatectomy procedures, but their original annotations are not designed for the task of surgical triplet detection. Despite the information presented in Table 1, which indicates that the ESAD dataset (Bawa et al., 2021) includes full bounding box annotations with 46,325 annotated instances, these annotations are largely unusable for precise surgical triplet detection.

The bounding boxes in ESAD are often coarsely defined, sometimes encompassing an entire instrument, and in other cases including multiple instruments and anatomical targets within a single box, as illustrated in Fig. 4. Consequently, we did not utilize ESAD's annotations and instead performed our own re-annotation to ensure precision and consistency for ProstaTD, and we applied the same unified annotation rules to our internally collected PWH dataset.

Similarly, the PSI-AVA dataset (Valderrama et al., 2022) provides 5,804 annotated instances with sparse bounding box annotations, which do not cover all frames. Although PSI-AVA includes over 5,000 bounding boxes, these annotations typically encompass the entire instrument, unlike our ProstaTD dataset, where the annotations focus specifically on the instrument's head

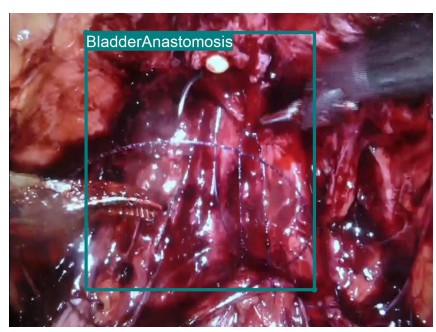

Figure 4: Coarse bounding box annotations in ESAD, showing examples where boxes include multiple instruments.

and mid-joint positions for greater precision. Therefore, for our 196,490 annotated instances in ProstaTD, we also opted to re-annotate rather than adopt PSI-AVA's bounding boxes to maintain consistency and alignment with our fine-grained annotation strategy.

### A.2    PROSTATD VIDEO PREPROCESSING

**Removal of Non-surgical Content**. The preprocessing stage entailed rigorous refinement of surgical videos sourced from our in-house PWH dataset, as well as the ESAD (Bawa et al., 2021) and PSI-AVA (Valderrama et al., 2022) datasets. The objective was to curate a high-quality dataset comprising only relevant surgical content. Initially, non-surgical frames, such as those captured during pre-operative preparation and post-operative recovery, were systematically removed. Additionally, intraoperative segments containing non-surgical visual content were excluded. These included frames with lens contamination necessitating endoscopic cleaning, instrument-switching-induced occlusions, and camera deviations from the surgical field for operational purposes.

**Quality-oriented Frame Filtering**. We also discarded frames with severely degraded visual quality, where even experienced surgeons could not reliably interpret surgical activity. Such instances included extreme lighting artifacts or dense surgical smoke that obscured the operative field. Retaining these frames would hinder effective annotation and risk undermining the dataset's overall reliability.

### A.3    PROSTATD INSTRUMENT TAXONOMY AND VARIATIONS

As discussed in the main text, surgical instruments exhibit significant visual variability across sources due to differences in manufacturer design and institutional preferences. Even within a single source, instruments of the same category may appear differently depending on their specific surgical applications (e.g., non–da Vinci systems). To illustrate the dataset's diversity, we present a comprehensive visualization of all instrument types in Fig. 5, capturing both inter-source variation

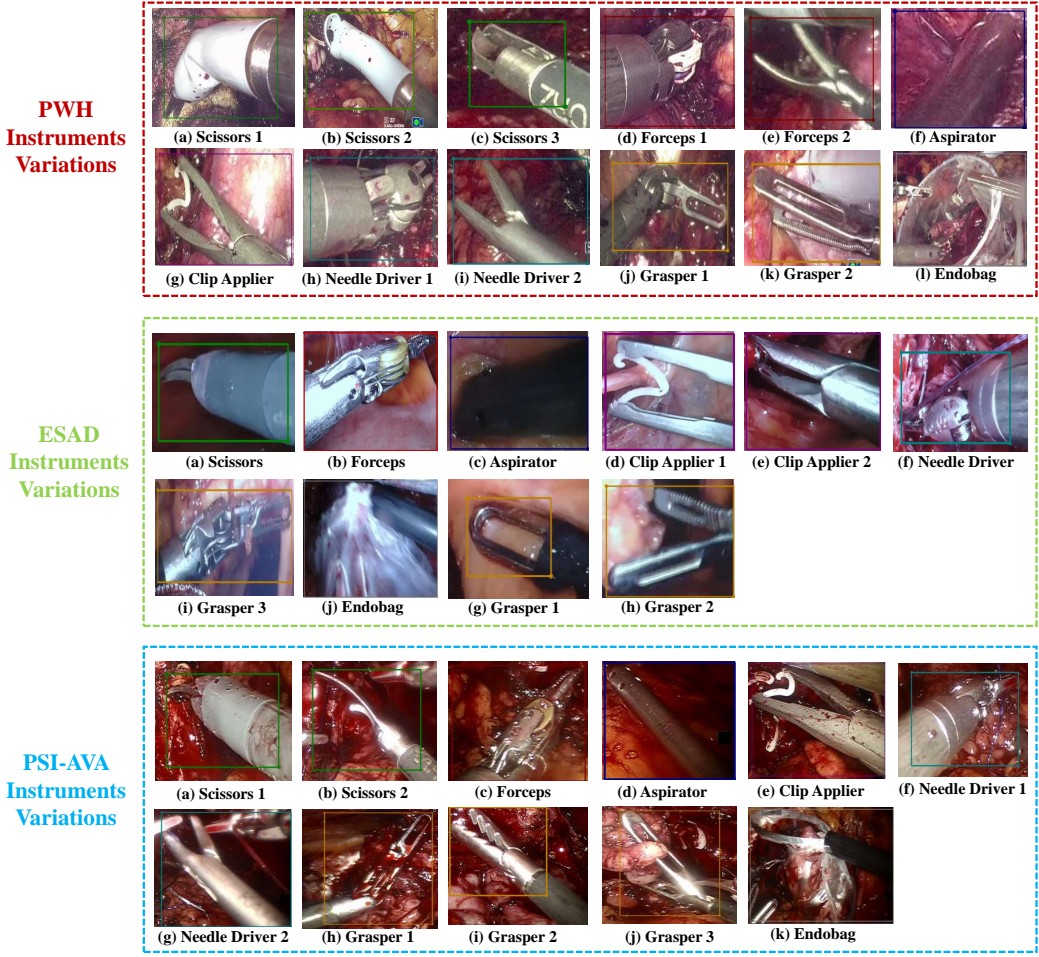

Figure 5: **Instrument appearance variation across datasets.** Representative examples with ground-truth bounding boxes from PWH *(red)*, ESAD *(green)*, and PSI-AVA *(blue)*. Beyond appearance inconsistencies across sources, some instrument *variants* are unique to specific datasets.

and intra-category differences. These examples also serve to intuitively demonstrate our bounding box annotation strategy across diverse instrument classes.

**Instrument Variations Across Datasets**. Our in-house PWH dataset comprises 12 distinct surgical instrument types, while the ESAD dataset (Bawa et al., 2021) includes 10 instrument types, and PSI-AVA (Valderrama et al., 2022) comprises 11. As depicted in Fig. 5, instruments from PWH (red bounding boxes) include frequently utilized da Vinci robotic instruments such as (a), (d), (f), (g), (h), (j), and (l) in prostatectomy procedures. A physically worn variant is shown in (b), reflecting real-world degradation, and a conventional laparoscopic device appears in (c), highlighting the inclusion of non-robotic instruments in minimally invasive procedures.

Instruments from ESAD (green bounding boxes) and PSI-AVA (blue bounding boxes) show partial overlaps, such as graspers (a in both), scissors (b in ESAD, c in PSI-AVA), and other shared types like (c), (d), (f), and (i) in ESAD, as well as (a), (c), (d), (e), (f), and (h) in PSI-AVA. These overlaps reflect a high degree of visual similarity, yet subtle design variations are evident, for instance, in the color details of the grasper (a) between ESAD and PSI-AVA. Additionally, unique instrument types appear in each dataset, such as (g) in ESAD and (b) in PSI-AVA. Compared across all sources, the instruments exhibit notable visual differences, emphasizing inter-dataset variability and intra-category diversity.

The integration of ESAD, PSI-AVA, and our PWH videos into the ProstaTD dataset introduces greater complexity compared to existing benchmarks such as CholecT50. The resulting diversity in

instrument appearances and variations offers a more realistic foundation for developing robust surgical instrument detection systems capable of generalizing across heterogeneous surgical environments and accommodating increasingly complex procedures.

### A.4 EXPERT-GUIDED LABEL REFINEMENT STRATEGY

Similar to prior work (Nwoye et al., 2022), we applied a systematic label consolidation strategy guided by two core principles: clinical relevance and semantic consistency. This step was essential, as raw surgical action labels often contain redundant or clinically marginal variations that may undermine the dataset's practical applicability.

The consolidation process was performed under clinical expert supervision and comprised two main stages. First, semantically equivalent action triplets were identified and merged into unified super-classes. This grouping operation, denoted as $\cup$, was carefully designed to preserve clinical meaning while reducing unnecessary label granularity. Representative examples of this merging strategy are provided in the following:

1. $\langle$grasper, retract, prostate$\rangle$ $\cup$ $\langle$grasper, retract, prostate-apex$\rangle$ $\cup$ $\langle$grasper, retract, DVC$\rangle$ $\longrightarrow$ $\langle$grasper, retract, prostate$\rangle$

2. $\langle$Endobag, bag, prostate$\rangle$ $\cup$ $\langle$Endobag, store, prostate$\rangle$ $\longrightarrow$ $\langle$Endobag, bag, prostate$\rangle$

3. $\langle$scissors, cut, vas-deferens$\rangle$ $\cup$ $\langle$scissors, cut, seminal-vesicle$\rangle$ $\longrightarrow$ $\langle$scissors, cut, seminal-vesicle$\rangle$

4. $\langle$forcep, retract, bladder$\rangle$ $\cup$ $\langle$forcep, retract, bladder-neck$\rangle$ $\longrightarrow$ $\langle$forcep, retract, bladder$\rangle$

5. $\langle$aspirator, suck, fluid$\rangle$ $\cup$ $\langle$aspirator, suck, smoke$\rangle$ $\cup$ $\langle$aspirator, suck, blood$\rangle$ $\longrightarrow$ $\langle$aspirator, suck, fluid$\rangle$

6. $\langle$scissors, dissect, fascia$\rangle$ $\cup$ $\langle$scissors, dissect, lymph node$\rangle$ $\cup$ $\langle$scissors, dissect, fat$\rangle$ $\longrightarrow$ $\langle$scissors, dissect, fascia$\rangle$

## B PROSTATD DATASET ANALYSIS AND PARTITIONING

### B.1 TRIPLET COMPONENT DISTRIBUTION IN PROSTATD DATASET

Table 6 reports triplet frequencies for ESAD, PSI, and PWH, together with overall totals, using standardized abbreviations for instruments, verbs, and targets. Each source dataset introduces subtle yet meaningful variations in instrument appearance, procedural conventions, and surgical technique, which collectively increase the diversity and difficulty of the learning task. This diversity is a deliberate design choice to promote generalization across real-world surgical scenarios. The distributional differences further justify integrating these datasets: ESAD exhibits markedly higher frequencies of fascias-related triplets (e.g., *scissors,dissect,fascias*); PSI shows distinctive patterns such as frequent vesicle- and aspirator-based retraction events (e.g., *aspirator,retract,vesicle*); PWH routinely includes Endobag-related contexts, leading to frequent bag-centric triplets (e.g., *bag,null,null*) as well as cases that are absent in the other datasets. Several triplet categories appear predominantly or even exclusively in a single dataset, demonstrating that each contributes unique, non-redundant information. Rather than a simple aggregation, their integration ensures broader coverage of clinical variability across institutions and techniques, which is essential for developing robust and generalizable models.

Beyond these distributional differences, ProstaTD also exhibits a pronounced long tail similar to CholecT50 (Nwoye et al., 2022), with many triplet categories appearing only a few times. This pattern is amplified by the multi-source composition and by variability in surgical styles, as different surgeons follow their own operational habits. For example, using forceps to perform a suture instead of a needle driver to save the time of switching instruments. Consequently, numerous infrequent triplets naturally arise. In real surgical settings, it is neither feasible to enumerate all possible instrument-action-target combinations nor to exclude human error or improvisation, so capturing this variability is necessary for building robust recognition systems. For approaches that decompose triplet prediction into separate instrument, action, and target branches, rare triplets provide valuable stress tests for generalization to uncommon scenarios, a capability that is critical for reliable deployment in the wild.

Table 6: Triplet frequencies by dataset (ESAD, PSI, PWH) with overall totals. Here, "driver" refers to "needle driver", "applier" to "clip applier", and "vesicle" abbreviates "seminal vesicle".

| Triplet | ESAD | PSI | PWH | Total | Triplet | ESAD | PSI | PWH | Total |
|---|---|---|---|---|---|---|---|---|---|
| scissors,retract,bladder | 26 | 359 | 481 | 866 | scissors,retract,catheter | 21 | 84 | 8 | 113 |
| scissors,retract,vesicle | 233 | 541 | 403 | 1177 | scissors,retract,prostate | 266 | 617 | 1326 | 2209 |
| scissors,retract,fascias | 646 | 233 | 278 | 1157 | scissors,retract,gauze | 0 | 0 | 92 | 92 |
| scissors,retract,Endobag | 0 | 3 | 99 | 102 | scissors,coagulate,bladder | 30 | 64 | 11 | 105 |
| scissors,coagulate,vesicle | 48 | 15 | 0 | 63 | scissors,coagulate,prostate | 36 | 7 | 0 | 43 |
| scissors,coagulate,fascias | 105 | 13 | 11 | 129 | scissors,cut,bladder | 529 | 2331 | 1356 | 4216 |
| scissors,cut,catheter | 0 | 4 | 0 | 4 | scissors,cut,vesicle | 80 | 470 | 192 | 742 |
| scissors,cut,prostate | 1608 | 3504 | 2290 | 7402 | scissors,cut,fascias | 280 | 14 | 163 | 457 |
| scissors,cut,thread | 18 | 43 | 31 | 92 | scissors,dissect,bladder | 375 | 176 | 74 | 625 |
| scissors,dissect,vesicle | 1177 | 1673 | 2767 | 5617 | scissors,dissect,prostate | 1348 | 1773 | 2100 | 5221 |
| scissors,dissect,fascias | 4734 | 750 | 1725 | 7209 | scissors,null,null | 4740 | 8221 | 6690 | 19651 |
| forceps,retract,bladder | 1448 | 5095 | 2696 | 9239 | forceps,retract,catheter | 0 | 2 | 33 | 35 |
| forceps,retract,vesicle | 1888 | 3696 | 3137 | 8721 | forceps,retract,prostate | 1704 | 4163 | 4444 | 10311 |
| forceps,retract,fascias | 4894 | 807 | 2595 | 8296 | forceps,retract,Endobag | 0 | 0 | 69 | 69 |
| forceps,coagulate,bladder | 0 | 413 | 267 | 680 | forceps,coagulate,vesicle | 0 | 115 | 403 | 518 |
| forceps,coagulate,prostate | 18 | 255 | 1201 | 1474 | forceps,coagulate,fascias | 341 | 88 | 158 | 587 |
| forceps,dissect,vesicle | 0 | 11 | 5 | 16 | forceps,dissect,prostate | 0 | 24 | 21 | 45 |
| forceps,dissect,fascias | 90 | 0 | 90 | 180 | forceps,grasp,catheter | 6 | 77 | 21 | 104 |
| forceps,grasp,vesicle | 63 | 0 | 0 | 63 | forceps,grasp,prostate | 9 | 17 | 46 | 72 |
| forceps,grasp,fascias | 443 | 10 | 36 | 489 | forceps,grasp,gauze | 0 | 60 | 116 | 176 |
| forceps,grasp,Endobag | 0 | 0 | 245 | 245 | forceps,grasp,thread | 764 | 32 | 782 | 1578 |
| forceps,suture,bladder | 38 | 0 | 85 | 123 | forceps,suture,prostate | 33 | 0 | 71 | 104 |
| forceps,suture,fascias | 15 | 0 | 0 | 15 | forceps,null,null | 3511 | 5331 | 4903 | 13745 |
| aspirator,retract,bladder | 395 | 4586 | 3438 | 8419 | aspirator,retract,vesicle | 14 | 461 | 0 | 475 |
| aspirator,retract,prostate | 52 | 248 | 964 | 1264 | aspirator,retract,fascias | 302 | 134 | 2324 | 2760 |
| aspirator,retract,Endobag | 0 | 0 | 13 | 13 | aspirator,suck,fluid | 4088 | 4345 | 5538 | 13971 |
| aspirator,null,null | 1167 | 1189 | 1477 | 3833 | driver,retract,bladder | 203 | 159 | 487 | 849 |
| driver,retract,prostate | 0 | 32 | 1 | 33 | driver,retract,fascias | 288 | 4 | 76 | 368 |
| driver,grasp,bladder | 0 | 0 | 5 | 5 | driver,grasp,catheter | 0 | 0 | 30 | 30 |
| driver,grasp,prostate | 0 | 0 | 18 | 18 | driver,grasp,fascias | 7 | 7 | 3 | 17 |
| driver,grasp,gauze | 0 | 0 | 79 | 79 | driver,grasp,Endobag | 0 | 0 | 57 | 57 |
| driver,grasp,thread | 1044 | 4233 | 3678 | 8955 | driver,suture,bladder | 315 | 382 | 1331 | 2028 |
| driver,suture,prostate | 117 | 277 | 173 | 567 | driver,suture,fascias | 35 | 23 | 0 | 58 |
| driver,null,null | 784 | 2017 | 2271 | 5072 | grasper,retract,bladder | 261 | 1484 | 228 | 1973 |
| grasper,retract,catheter | 0 | 36 | 25 | 61 | grasper,retract,vesicle | 461 | 3074 | 5251 | 8786 |
| grasper,retract,prostate | 270 | 2466 | 3081 | 5817 | grasper,retract,fascias | 647 | 494 | 1079 | 2220 |
| grasper,grasp,catheter | 16 | 88 | 21 | 125 | grasper,grasp,vesicle | 21 | 6 | 29 | 56 |
| grasper,grasp,prostate | 85 | 127 | 172 | 384 | grasper,grasp,fascias | 193 | 0 | 138 | 331 |
| grasper,grasp,gauze | 0 | 9 | 32 | 41 | grasper,grasp,Endobag | 4 | 70 | 362 | 436 |
| grasper,grasp,thread | 0 | 89 | 0 | 89 | grasper,null,null | 649 | 2225 | 2904 | 5778 |
| applier,clip,bladder | 19 | 87 | 37 | 143 | applier,clip,vesicle | 56 | 211 | 24 | 291 |
| applier,clip,prostate | 100 | 478 | 296 | 874 | applier,clip,fascias | 135 | 8 | 286 | 429 |
| applier,clip,Endobag | 0 | 0 | 39 | 39 | applier,null,null | 660 | 749 | 883 | 2292 |
| Endobag,bag,prostate | 54 | 265 | 451 | 770 | Endobag,bag,fascias | 0 | 0 | 127 | 127 |
| Endobag,null,null | 54 | 96 | 2230 | 2380 | **Total:** | **44061** | **71250** | **81179** | **196490** |

## B.2 FIVE-FOLD CROSS-VALIDATION DATASET PARTITION PROTOCOL

Table 7: Five-fold cross-validation split for our experimental setup.

| Fold | Test Videos | | | | | Validation Videos | | Training Videos |
|---|---|---|---|---|---|---|---|---|
| 1 | esadv1 | psiv1 | psiv4 | pwhv8 | | psiv7 | pwhv5 | Remaining videos |
| 2 | esadv2 | psiv7 | pwhv4 | pwhv9 | | psiv1 | pwhv6 | Remaining videos |
| 3 | esadv3 | psiv14 | pwhv1 | psiv2 | | esadv1 | pwhv2 | Remaining videos |
| 4 | esadv4 | psiv15 | pwhv2 | pwhv7 | | esadv3 | pwhv3 | Remaining videos |
| 5 | psiv3 | psiv21 | pwhv3 | pwhv5 | pwhv6 | psiv14 | pwhv1 | Remaining videos |

In our experiments, we adopt a five-fold cross-validation scheme over all 21 surgical videos, rotating the test set so that every video (and the rare triplets it contains) appears in testing at least once. For each fold, the remaining videos form the pool for training and validation. In our default protocol, two videos are randomly selected from this pool as the validation set, and the rest are used for training. The validation selections given in Table 7 can be used verbatim to replicate our results. Alternatively, researchers may adopt a different protocol that omits the validation set. This is because validation sets often inadvertently contain rare triplets, which can lead to significant performance bias. This issue was also observed in the CholecT50 dataset. While we adhere to the standard validation split to

remain consistent with prior work, skipping the validation set is a recommended approach to address data scarcity.

## C    ANNOTATION SOFTWARE

To our knowledge, before this work there was no open-source annotation tool purpose-built for surgical instance, and none that supports structured triplet labels. In existing general annotation tools, annotators must select each instance and then manually assign one class out of 89 categories, which makes the process slow and error-prone. Such a workflow is impractical for large-scale triplet annotation in surgical videos because it scales poorly across frames, instruments, actions and targets. Even with a large team, using general tools would take years to produce only a small portion of the required labels.

To construct ProstaTD at scale we developed dedicated software for triplet labeling *(instrument, action, target, bbox)*. Although the initial motivation came from our prostatectomy dataset, the design is procedure agnostic and transfers to other surgeries with different organ targets and workflow conventions. This provides a practical step toward a general-purpose foundation model for surgical annotation. We introduce two complementary tools. **Triplet-labelme** supports single-frame editing, allowing annotators to assign triplet attributes *(instrument, action, target)* to every instance in an image and to draw or refine the corresponding bounding box for each instance. **SurgLabel** supports batch annotation by using existing bounding boxes and instance identities together with an adaptive track identity assignment, which propagates and verifies action and target labels across the timeline. Together these tools address diverse needs in surgical video analysis.

### C.1    TRIPLET-LABELME ANNOTATION TOOL

This tool is adapted from LabelMe[1] and optimized for surgical use (see Fig. 6). It streamlines bounding box creation and instrument category assignment, and supports single-frame editing of triplet annotations *(instrument, action, target)* for every instance in an image together with the corresponding bounding box. Users can freely modify the action and the target of any selected instance.

While primarily designed for triplet annotation in ProstaTD, the tool is procedure agnostic and can be applied to other surgical tasks such as per-image segmentation, surgical phase and step annotation, or higher-level grouping labels. As shown in Fig. 6, users can import a custom JSON file to give each instance the attributes *action*, *target*, and *track identity*. The detailed JSON file and three custom examples are described in Section H.1. Specifically, by modifying the imported JSON, users can restrict the permitted action and target values for a given instrument category to a specified choice set, and can also extend the imported JSON schema to add custom attributes beyond action and target.

Productivity and stability are enhanced through default auto-save, ergonomic shortcut mappings for mode switching, for example changing key "Ctrl+J" to simply key "J", and flexible visualization controls for colors, label transparency, font size, and line thickness. Visualization label boxes are adaptively positioned to avoid overlapping, which keeps the display clear. In addition, we fixed several LabelMe issues that previously caused crashes or made some operations ineffective during large-scale annotation.

### C.2    SURGLABEL ANNOTATION TOOL

After completing instrument instance annotation with Triplet-Labelme, the next step is to assign actions and targets to these instances. Unlike instrument annotation, which can be partially assisted by our trained models, action and target labeling is temporally extended and context-dependent, and becomes inefficient if performed frame by frame. To avoid slow per-frame editing and to improve temporal consistency, we designed **SurgLabel** (see Fig. 7), a span-based **batch annotation** interface purpose-built for surgical videos. It treats temporal spans as first-class objects, supports on-the-fly labeling while scrubbing the timeline or scrolling through the dataset, and integrates seamlessly with the instrument instances defined in Triplet-Labelme. Its main capabilities are as follows:

---

[1]https://github.com/wkentaro/labelme

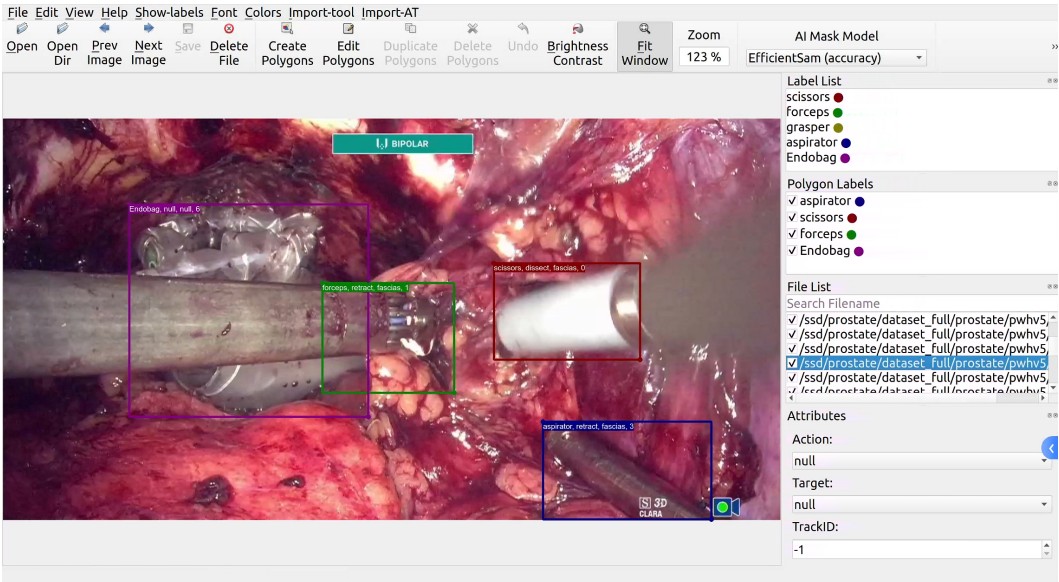

Figure 6: Our customized **Triplet-Labelme** tool for bounding box and instrument category annotation with per-frame triplet editing *(instrument, action, target)*.

**Span-based batch labeling.** Select one or multiple instrument instances and specify start and end frame indices to apply action and target labels over a contiguous temporal range in a single operation, which enables on-the-fly annotation during scrubbing or playback and keeps long maneuvers consistent with minimal interactions.

**Open action/target schema.** SurgLabel supports arbitrary action and target sets by importing a user-defined JSON label schema that specifies, for each instrument class, its admissible actions and targets (see Section H.1 for details and examples). The same schema can be reused across procedures, so teams can standardize or switch configurations without code changes.

**Support for Diverse Surgical Tasks.** The same span-based mechanism applies to image-level annotations such as surgical phase and step, and to instance-level annotations such as orientation and free-form descriptions (see Section H.1 for details). As a result, SurgLabel is not limited to surgical triplets and can serve diverse labeling tasks, supporting broader data curation for future foundation models.

**Support segmentation instances.** Although SurgLabel was initially designed for bounding box workflows, it also supports segmentation. Action and target labels, as well as any schema attributes, can be attached in batch to segmentation *instances*, and the interface mirrors the box mode (see Fig. 7). This keeps geometric masks and semantic triplets synchronized when instruments persist across spans.

**Adaptive track identity assignment.** SurgLabel provides automatic per-class instance numbering that remains stable when multiple instruments of the same type are present. By default, it relies on spatial and motion heuristics guided by surgical priors: instruments are inserted through trocars, operate from a relatively fixed direction in the camera view, and stay in that sector until fully removed. We then compare instruments of the same class hierarchically using image region, orientation, and keypoint coordinates to separate instances over time. This rule is user-configurable, and switching strategies is possible when situations change. The resulting assignment is smoothed temporally, so brief crossings or short occlusions rarely cause identity swaps. This resolves most trocar ingress and egress cases, as shown in Fig. 7. Users may also switch to the ByteTrack (Zhang et al., 2022) method via the **Sort Method** button, but we recommend the default because it is more stable, especially under low-frame-rate annotation (e.g., 1 fps). Assigned IDs are temporary and the sort method can be modified at any time, and same-class instruments are displayed as #1, #2, #3 for unambiguous labeling. Overall, track ID annotation is time-consuming, particularly in non-robotic procedures. Leveraging this prior knowledge-based automatic assignment greatly reduces the workload. Alternatively, if track

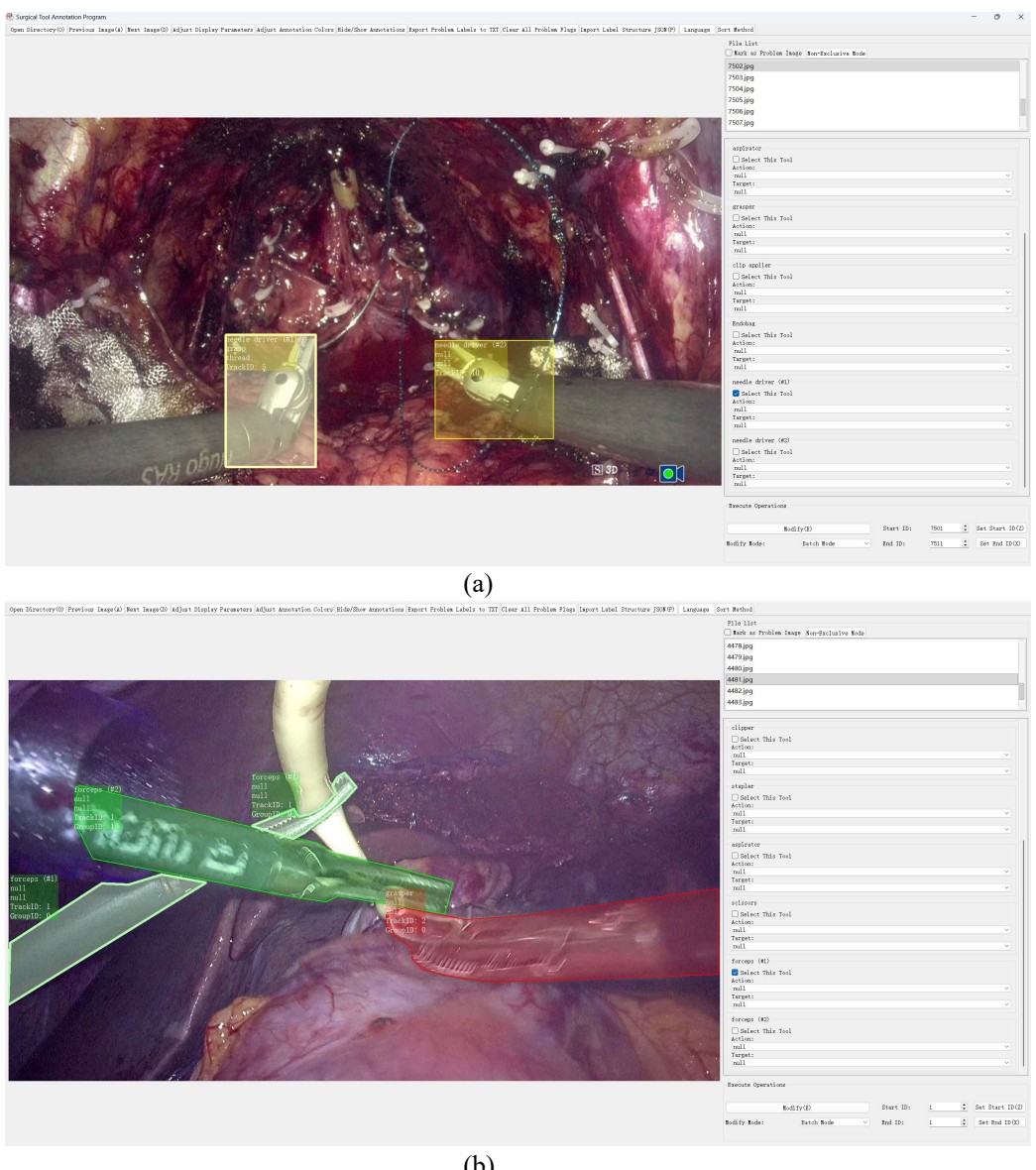

Figure 7: **SurgLabel**, a purpose-built interface for span-based batch labeling of actions and targets with predefined instrument instances. It supports automatic disambiguation of concurrent instruments of the same category by assigning persistent instance tags across time (e.g., #1, #2), ensuring temporal consistency. (a) Detection mode, (b) Segmentation mode.

IDs are already available upstream, our SurgLabel can also directly support batch annotation from them without re-assignment.

**User-friendly ergonomics.** The interface provides multilingual support, ergonomic shortcuts, selection highlighting, and autosave to reduce friction at scale. Overlays use adaptive transparency so labels do not occlude important content. Different visualization label boxes are adaptively positioned to avoid overlapping, ensuring clear visibility. Annotators can navigate frames with the **A** and **D** keys, set the start and end frames with the **Z** and **X** keys, and confirm edits with the **E** key, which streamlines repetitive operations.

**Rich customization.** Per-class colors, label and box opacity, highlight intensity, line thickness, and font weight are configurable via the **Adjust Display Parameters** button and the **Adjust Annota-**

**tion Colors** button, allowing teams to enforce consistent visual conventions while accommodating individual preferences. For more details, please see Appendix.H.2.

**Open sourcing.** We will release both tools as open source together with documentation, configuration templates, and example projects, which will facilitate annotation across diverse surgical tasks and accelerate community progress toward surgical foundation models.

## D  EVALUATION TOOLKIT

To better evaluate surgical triplet detection performance, building upon the prior "ivtmetrics" work in CholecT50 (Nwoye & Padoy, 2022), we have developed an enhanced evaluation toolkit named "ivtdmetrics." Compared to the original "ivtmetrics," our toolkit introduces the following key enhancements:

---

**Key Evaluation Enhancements**

(1) **Global Confidence Ranking**: implemented global confidence score ranking for mAP calculation instead of image-level ranking, ensuring consistent evaluation across all frames. This follows the COCO evaluation protocol, where predictions across the entire dataset are ranked globally.

(2) **101-Point Interpolation**: adopted 101-point interpolation for mAP calculation to align with COCO-style evaluation standards.

(3) **Pseudo Detection Handling**: fixed calculation errors when handling pseudo detections for scenarios where ground truth lacks certain classes but predictions include them.

(4) **Precision, Recall, and F1-score Evaluation**: added metrics based on a single optimal confidence threshold determined by maximizing the F1 score.

(5) **mAP@50:95 Evaluation**: added mAP@50:95 result calculation, averaging over IoU thresholds from 0.5 to 0.95.

(6) **Support for Verb and Target**: added support for verb (V) and target (T) component evaluations alongside instrument (I) and full instrument-verb-target (IVT).

(7) **Video-wise Calculation**: added support for video-wise metric calculation across all components, ensuring each surgical procedure contributes equally regardless of its duration.

(8) **Bug Fixes**: corrected several issues, including errors in the "list2stack" function from the prior work, thereby improving robustness.

---

More specifically, the "ivtdmetrics" toolkit integrates common detection metrics into a unified package, enabling simultaneous computation across instruments (I), verbs (V), targets (T), and full instrument–verb–target triplets (IVT). It supports mean Average Precision at a fixed IoU threshold of 0.5 (mAP@50) and averaged over IoU thresholds from 0.5 to 0.95 in steps of 0.05 (mAP@50:95).

Formally, for each component $X \in \{I, V, T, IVT\}$ and class $c$, the Average Precision is defined as

$$\mathrm{AP}_c^X = \int_0^1 p_{\mathrm{interp}}^X(r)\,dr \approx \mathrm{trapz}\big(p_{\mathrm{interp}}^X(x),\, x\big), \quad x = \mathrm{linspace}(0, 1, 101), \qquad (1)$$

where $p_{\mathrm{interp}}^X(r)$ denotes the interpolated precision envelope as a function of recall, and $\mathrm{trapz}$ indicates trapezoidal numerical integration over 101 uniformly spaced recall points.

The mean Average Precision (mAP) for each component is then

$$\mathrm{mAP}_X = \frac{1}{C_X} \sum_{c=1}^{C_X} \mathrm{AP}_c^X, \quad X \in \{I, V, T, IVT\}, \qquad (2)$$

where $C_X$ is the number of valid classes for component $X$.

Moreover, Precision, Recall, and F1 score are computed at a single optimal confidence threshold determined by maximizing the global F1 score across all classes, and results are reported in percentages with higher values indicating better performance ($\uparrow$).

**Video-wise mAP.** Besides the global evaluation, we also report video-wise metrics. Formally, let $\text{AP}_c^X(v)$ denote the Average Precision of component $X \in \{I, V, T, IVT\}$ and class $c$ computed on video $v$. Given $V$ videos, the video-wise mAP is defined as

$$\text{mAP}_X^{\text{video}} = \frac{1}{V} \sum_{v=1}^{V} \frac{1}{C_X} \sum_{c=1}^{C_X} \text{AP}_c^X(v), \tag{3}$$

where $C_X$ is the number of valid classes for component $X$. In this setting, confidence scores are ranked within each video rather than globally, ensuring that the evaluation is self-contained per procedure. This protocol ensures that each video contributes equally, mitigating bias from variable procedure lengths and the absence of certain rare triplets in different five-fold splits.

Our "ivtdmetrics" toolkit is publicly available for reproducibility, with detailed implementation and usage examples provided in the accompanying code repository.

# E   OUR PROPOSED TDNET NETWORK

Following the progress of surgical triplet approaches (Yamlahi et al., 2023; Gui et al., 2023) in classification tasks, we propose a tailored baseline method named TDnet to support future research on surgical triplet detection. This method is designed to tackle the severe class imbalance issue in triplet annotations and to provide a robust baseline for subsequent studies. In the following, we introduce TDnet from two complementary aspects: the network architecture and the overall loss function.

## E.1   NETWORK ARCHITECTURE

As shown in Table 6, our surgical triplet dataset exhibits a severe distribution imbalance across categories. We therefore propose a distillation-based baseline for surgical triplet detection on ProstaTD to facilitate future comparisons. Self-distillation supplies softened targets that smooth the label distribution and encode similarity between classes, which reduces overconfidence on frequent categories and improves calibration for rare ones. We are the first to apply this mechanism at the *instance level*, tying supervision to each positive detection rather than global image labels. This alignment with the detection objective provides more localized gradients and proves more effective in our setting, especially for rare triplets. In addition, the network jointly integrates auxiliary classification heads for instrument, action, and target at the instance level under the supervision of both hard and soft labels, where the hard labels provide exact supervision to the ground truth while the softened teacher logits act as a regularizer that captures ambiguity and proximity between classes. Mixing the two improves generalization on underrepresented triplets without sacrificing fidelity to the annotations.

The overall architecture of our proposed TDnet is illustrated in Fig. 8. It consists of a teacher network and a student network, both built upon the same YOLOv12 backbone and detection head. The teacher is first trained to predict triplet detections together with three auxiliary heads for instrument, action, and target. Instead of running separate detection sample assignment for each auxiliary head, we reuse the positive bounding box predictions selected by the triplet head. Specifically, the main head selects top-$k$ bounding box predictions per ground truth triplet using a score- and IoU-based criterion to form a foreground set $\mathcal{S}$. The same foreground set is then applied to the auxiliary heads, and the corresponding instrument, action, and target labels from the matched triplet are propagated to these positives. Auxiliary losses are computed with binary cross entropy on $\mathcal{S}$ while the remaining grid points are treated as background. This shared assignment avoids inconsistent supervision across heads and stabilizes training.

Formally, let $g \in \mathcal{G}$ denote a ground truth triplet with box $b_g$ and class $c_g$, and let $a \in \mathcal{A}$ denote a box prediction at a grid point, with predicted box $b_a$ and triplet head logit $z_a(c_g)$. We define the alignment score:

$$s(a, g) \;=\; \sigma\big(z_a(c_g)\big)^{\alpha} \cdot \text{IoU}\big(b_a, b_g\big)^{\beta}, \tag{4}$$

where $\sigma(\cdot)$ is the sigmoid function and $\alpha, \beta \geq 0$ are hyperparameters. For each $g$, we select the top-$k$ bounding box predictions by this score:

$$\mathcal{S}_g \;=\; \text{TopK}_k \big\{ s(a, g) \,\big|\, a \in \mathcal{A} \big\}, \tag{5}$$

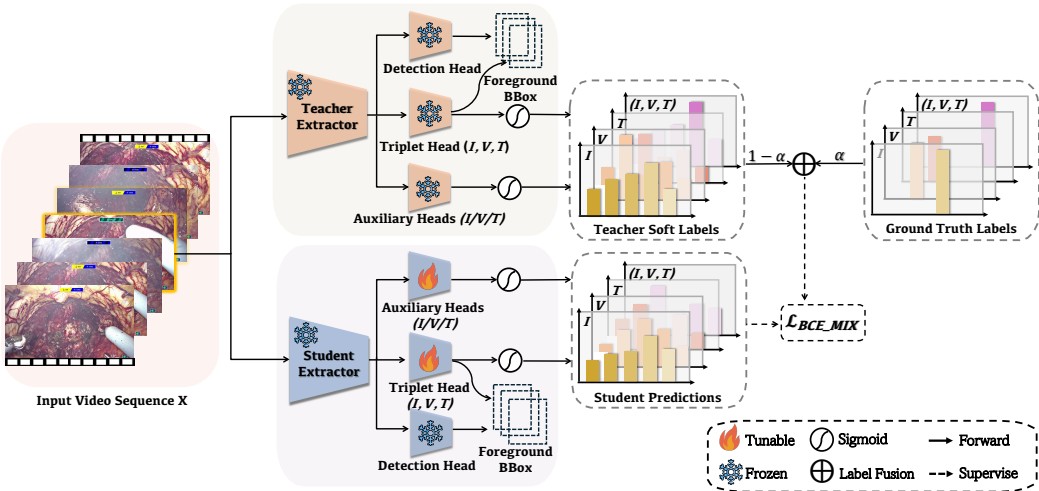

Figure 8: Overview of TDnet. A teacher and a student share a YOLOv12-style feature extractor. First, the teacher network is trained for triplet detection. During this stage, positive samples selected by the main triplet head define a shared foreground mask, which is used to supervise the auxiliary instrument, action, and target heads. In the subsequent self-distillation phase, the frozen teacher provides softened logits. The student then optimizes all four of its classification heads on this same shared foreground using a fused BCE objective. This approach mitigates class imbalance and enhances multi-task performance.

and form the shared foreground set:

$$\mathcal{S} = \bigcup_{g \in \mathcal{G}} \mathcal{S}_g, \qquad g^\star(a) = \arg\max_{g \in \mathcal{G}} s(a, g) \text{ for predictions selected by multiple } \mathcal{S}_g. \quad (6)$$

The labels for the auxiliary heads are inherited from $c_{g^\star(a)}$ and losses are computed only on $a \in \mathcal{S}$. In all our experiments, we set $\alpha = 0.5$, $\beta = 6.0$, and $k = 10$.

In the subsequent self-distillation stage, the teacher network is frozen and produces softened logits for the four classification branches: triplet, instrument, action, and target. The student copies the teacher backbone and detection head and keeps them frozen so that bounding box prediction geometry and responses remain identical to the teacher. This preserves the positive set selected by the triplet head, and we reuse this shared foreground for all branches during self distillation. Since box regression is not updated in this stage, supervision is purely classification on the shared positives. Each student classification head is trained with a fused binary cross entropy objective that blends the teacher probabilities with the original labels. Using softened targets reduces the dominance of frequent classes and improves calibration, and focusing the loss on the shared positives avoids the large negative-to-positive imbalance. Mixing teacher predictions with ground truth stabilizes supervision for rare triplets and yields consistent gains across instrument, action, target, and triplet heads.

### E.2 Loss Functions

**Teacher Network.** We reuse the shared positive set $\mathcal{S}$ defined in Section E.1, and let $\mathcal{A}$ denote all box predictions at grid points across the three feature levels. Here IVT denotes the joint triplet head that predicts the structured instrument, verb, and target combination, and I, V, T denote the auxiliary heads for instrument, verb, and target, respectively. The teacher is trained with a unified objective that applies box and DFL losses on positives, uses BCE on all grid-point predictions for the triplet head, and uses BCE on positives only for the auxiliary heads. Formally, the teacher loss is:

$$\underbrace{\sum_{a \in \mathcal{S}} \left( \lambda_{\text{box}} \mathcal{L}_{\text{box}}(a) + \lambda_{\text{DFL}} \mathcal{L}_{\text{DFL}}(a) \right)}_{\text{box and DFL on positives}} + \underbrace{\alpha_{\text{IVT}} \sum_{a \in \mathcal{A}} \mathcal{L}_{\text{BCE}}^{\text{IVT}}(a)}_{\text{triplet BCE on all predictions}} + \underbrace{\sum_{h \in \{\text{I,V,T}\}} \alpha_h \sum_{a \in \mathcal{S}} \mathcal{L}_{\text{BCE}}^{h}(a)}_{\text{auxiliary BCE on positives}}. \quad (7)$$

**Student Network.** In the self distillation stage the teacher is frozen and produces softened logits for the four heads. The student backbone and detection head are copied from the teacher and kept frozen so that the positive set $\mathcal{S}$ is identical. Distillation is classification only and is computed on $a \in \mathcal{S}$ for all heads using a fused BCE objective:

$$\mathcal{L}_{\text{student}} = \sum_{h \in \{\text{IVT,I,V,T}\}} \sum_{a \in \mathcal{S}} \mathcal{L}^h_{\text{BCE-MIX}}(a), \tag{8}$$

where the mixed soft target is

$$t^h(a) = \alpha\,\sigma\big(z_t^h(a)\big) + (1-\alpha)\,y^h(a), \qquad \alpha \in [0,1], \tag{9}$$

and

$$\mathcal{L}^h_{\text{BCE-MIX}}(a) = \text{BCE}\big(z_s^h(a),\, t^h(a)\big). \tag{10}$$

Here $z_t^h(a)$ and $z_s^h(a)$ are teacher and student logits for head $h$, $\sigma(\cdot)$ is the sigmoid function, and $y^h(a)$ is the original target. Softened teacher logits reduce the dominance of frequent classes and improve calibration, and focusing supervision on $\mathcal{S}$ avoids the extreme negative-to-positive imbalance while preserving consistent alignment between teacher and student.

## F  EXPERIMENT DETAILS

### F.1  IMPLEMENTATION DETAILS.

All experiments in Section 6 and Appendix F use an input size of $640 \times 640$ and are trained on an NVIDIA RTX 4090 GPU. For lightweight detectors such as the YOLO series, we use a batch size of 16, whereas larger models like DETR-based architectures are trained with smaller batch sizes (3–4) to fit memory constraints. The learning rate and warmup schedule are adapted according to the corresponding extractor.

For our TDnet, we use an input size of 640×640 and a batch size of 16. We set $\lambda_{\text{box}} = 7.5$, $\lambda_{\text{DFL}} = 1.5$, and $\alpha_h = 0.5$ for all classification heads. The classification loss weights are 0.5 for the IVT head and 0.1 for each of the I, V, and T heads. During self distillation, the mixed target assigns 0.8 to the teacher probabilities and 0.2 to the original labels for our triplet recognition. Data augmentation includes horizontal flipping with probability 0.5, random scaling by up to 50%, random translation up to 10% of the image size, mosaic augmentation with probability 1.0, and HSV color jitter consisting of hue variation of 0.015, saturation scaling up to 0.7, and value scaling up to 0.4. Optimization is performed using SGD with an initial learning rate of 0.0001, momentum of 0.937, and weight decay of $5 \times 10^{-4}$. The early-stopping patience is set to 100 epochs. Detailed hyperparameter configurations are available in the released configuration files and training scripts in our repository.

### F.2  VIDEO-WISE COMPARISON RESULTS

To provide a more comprehensive assessment, we conducted an additional **video-wise** mAP evaluation, whose definition is given in Appendix D. This protocol computes metrics per video and then averages across videos, which reduces bias from variation in procedure length and ensures that each case receives equal weight.

As shown in Table 8, the overall ranking is consistent with Table 4. Classical detectors such as Faster RCNN, Cascade RCNN, SSD, ViT-Det, and Deformable-DETR remain clearly behind. Task-specific TAPIR and MCIT-IG also underperform. Advanced detectors including RT-DETR (Zhao et al., 2024) and the YOLO family maintain strong results.

Our TDnet retains the top position, and its advantage is more pronounced under the **video-wise** metric on the critical triplet score. Concretely, TDnet leads on the other three components ($\text{mAP}_{\text{I}}$, $\text{mAP}_{\text{V}}$, and $\text{mAP}_{\text{T}}$), and on the primary triplet metric it surpasses the runner-up YOLOv12 by **2.4%** at $\text{mAP}_{\text{IVT}}$@0.5 and **2.1%** at $\text{mAP}_{\text{IVT}}$@0.50:0.95, exceeding the corresponding margins in Table 4 which are **1.8%** and **1.3%**.

### F.3  ABLATION STUDY

**Effect of Multi-task Learning and Self Distillation.** TDnet without self distillation, that is TDnet with only multi-task learning, shows the largest degradation relative to TDnet, most clearly on $\text{mAP}_{\text{V}}$

Table 8: Video-wise detection performance on ProstaTD. Metrics are computed per video and then averaged across videos. Results are reported as mean$_{\pm\text{std}}$ % over 5-fold cross-validation. We report video-wise mAP at IoU thresholds (50 and 50:95) for I, V, T, and IVT components. All metrics are in % with higher values indicating better performance (↑). Bold text with light green background indicates the best result, and underlined text with light purple background indicates the second best.

| Method | $mAP_I$ (%) ↑ | | $mAP_V$ (%) ↑ | | $mAP_T$ (%) ↑ | | $mAP_{IVT}$ (%) ↑ | |
|---|---|---|---|---|---|---|---|---|
| | 50 | 95 | 50 | 95 | 50 | 95 | 50 | 95 |
| Faster RCNN | $72.2_{5.1}$ | $62.7_{5.7}$ | $47.5_{5.5}$ | $41.6_{5.3}$ | $40.6_{4.5}$ | $35.0_{4.1}$ | $25.1_{4.0}$ | $21.7_{3.8}$ |
| Cascade RCNN | $68.6_{5.3}$ | $59.0_{5.9}$ | $43.7_{5.7}$ | $38.0_{5.4}$ | $38.6_{4.7}$ | $33.1_{4.3}$ | $21.1_{4.1}$ | $18.2_{3.9}$ |
| SSD | $73.6_{4.9}$ | $63.9_{5.5}$ | $49.4_{5.4}$ | $43.2_{5.2}$ | $42.4_{4.4}$ | $36.9_{4.0}$ | $26.4_{3.9}$ | $23.2_{3.6}$ |
| Vit-Det | $85.9_{2.3}$ | $73.0_{2.3}$ | $51.6_{3.8}$ | $44.7_{3.3}$ | $46.7_{4.3}$ | $40.1_{3.3}$ | $29.4_{3.5}$ | $25.9_{3.1}$ |
| Deformable-DETR | $74.3_{5.0}$ | $64.1_{5.5}$ | $49.9_{5.4}$ | $43.6_{5.2}$ | $42.5_{4.4}$ | $37.1_{4.0}$ | $25.8_{3.9}$ | $22.5_{3.6}$ |
| RT-DETR | $91.4_{1.6}$ | $80.9_{1.9}$ | $60.4_{3.8}$ | $54.4_{3.3}$ | $52.1_{4.4}$ | $46.5_{3.8}$ | $32.8_{3.8}$ | $29.3_{3.3}$ |
| YOLOv10 | $88.6_{1.5}$ | $80.8_{2.4}$ | $61.5_{2.5}$ | $56.8_{2.6}$ | $50.6_{2.6}$ | $46.8_{2.3}$ | $34.0_{3.3}$ | $31.5_{2.8}$ |
| YOLOv11 | $88.2_{1.3}$ | $80.7_{1.6}$ | $61.7_{3.0}$ | $56.8_{2.9}$ | $51.8_{2.3}$ | $47.7_{2.7}$ | $34.5_{2.6}$ | $31.7_{2.6}$ |
| YOLOv12 | $88.8_{1.2}$ | $80.5_{2.2}$ | $61.8_{2.6}$ | $56.6_{2.9}$ | $51.4_{2.6}$ | $46.9_{2.2}$ | $34.7_{3.6}$ | $31.8_{3.4}$ |
| TAPIR | $75.6_{4.4}$ | $65.2_{4.6}$ | $51.8_{4.9}$ | $45.0_{4.4}$ | $46.5_{5.1}$ | $39.9_{4.5}$ | $27.8_{4.4}$ | $24.1_{3.9}$ |
| MCIT-IG | $76.9_{4.1}$ | $66.6_{4.5}$ | $53.0_{4.8}$ | $46.3_{4.1}$ | $47.8_{5.0}$ | $41.2_{4.4}$ | $29.1_{4.2}$ | $25.4_{3.8}$ |
| TDnet (Ours) | $90.1_{1.8}$ | $81.2_{2.2}$ | $63.7_{2.3}$ | $58.3_{2.7}$ | $53.1_{2.3}$ | $48.5_{2.1}$ | $37.1_{2.9}$ | $33.9_{2.9}$ |

Table 9: Ablation results for TDnet. MTL denotes multi-task learning across instrument, action and target. The variant without self distillation uses only MTL. The variant without MTL uses only self distillation. TDnet uses both components. We report global mAP for instrument I, verb V, target T and triplet IVT at IoU 50 and 95. Bold text with light green background indicates the best in each column. The mAP values are computed in exactly the same way as in Table 4 in Section 6.3.

| Method | $mAP_I$ | | $mAP_V$ | | $mAP_T$ | | $mAP_{IVT}$ | |
|---|---|---|---|---|---|---|---|---|
| | 50 | 95 | 50 | 95 | 50 | 95 | 50 | 95 |
| TDnet w/o self-distill | $88.5_{1.3}$ | $80.5_{2.2}$ | $59.7_{3.3}$ | $55.0_{2.4}$ | $54.6_{3.4}$ | $50.7_{2.0}$ | $34.4_{4.2}$ | $31.8_{3.6}$ |
| TDnet w/o MTL | $89.5_{1.2}$ | $80.9_{2.0}$ | $60.9_{3.0}$ | $55.7_{2.1}$ | $55.5_{2.1}$ | $50.7_{2.0}$ | $35.4_{3.7}$ | $32.6_{3.2}$ |
| TDnet | $89.9_{1.3}$ | $81.0_{2.0}$ | $61.7_{2.9}$ | $56.3_{2.1}$ | $55.7_{2.4}$ | $50.8_{2.7}$ | $36.1_{3.5}$ | $33.1_{3.1}$ |

and mAP$_{IVT}$. Concretely, mAP$_V$@0.5 drops from 61.7% to 59.7% and mAP$_V$@0.95 drops from 56.3% to 55.0%. mAP$_{IVT}$@0.5 drops from 36.1% to 34.4% and mAP$_{IVT}$@0.95 drops from 33.1% to 31.8%, with smaller declines on mAP$_I$ and mAP$_T$. This suggests that directly optimizing three coupled heads on a shared feature extractor without auxiliary guidance makes the objective overly complex, increases interference among instrument, action, and target, and weakens the discriminative capacity of the learned representation.

TDnet without multi-task learning, that is TDnet with only self distillation, underperforms the full TDnet by a smaller margin. mAP$_V$@0.5 decreases to 60.9% and mAP$_V$@0.95 to 55.7%. mAP$_{IVT}$@0.5 decreases to 35.4% and mAP$_{IVT}$@0.95 to 32.6%, again with minor drops on mAP$_I$ and mAP$_T$. During distillation we align teacher and student detection boxes, which requires freezing the student feature extractor and the detection head so that the geometry remains stable. This constraint limits feature adaptation and therefore constrains the benefit of self distillation alone.

The full TDnet that combines multi-task learning with self distillation achieves the best values in every column of Table 9. Distillation stabilizes features and reduces cross-task conflict, while multi-task losses provide complementary supervision for instrument, action, and target. Although the feature extractor and the detection head are frozen during the box alignment stage, the remaining modules continue to learn jointly, which yields consistent gains, especially on mAP$_V$ and mAP$_{IVT}$ where cross-task dependencies are strongest.

**Effect of TDnet MTL head on different extractors.** We replace the YOLOv12 backbone in TDnet with YOLOv10 and YOLOv11 extractors and measure IVT performance under five-fold cross-validation. Results are summarized in Table 10.

Table 10: Impact of inserting the TDnet MTL head on different extractors (five-fold cross-validation). We report IVT-level mAP at IoU 50 and 95.

| Method | $\text{mAP}_{\text{IVT},50}$ | $\text{mAP}_{\text{IVT},95}$ |
|---|---|---|
| YOLOv10 | $34.3_{4.1}$ | $31.8_{3.5}$ |
| TDnet with YOLOv10 extractor | $35.1_{3.9}$ | $32.6_{3.3}$ |
| YOLOv11 | $34.1_{3.7}$ | $31.5_{3.3}$ |
| TDnet with YOLOv11 extractor | $34.9_{3.5}$ | $32.3_{3.1}$ |

Table 11: Sensitivity of TDnet to MTL–self-distillation fusion weights on a single fold. We report IVT-level mAP at IoU 50 and 95.

| Fusion weights ($\lambda_{\text{MTL}} : \lambda_{\text{SD}}$) | $\text{mAP}_{\text{IVT},50}$ | $\text{mAP}_{\text{IVT},95}$ |
|---|---|---|
| 0.0 : 1.0 | 40.9 | 37.8 |
| 0.2 : 0.8 | 40.8 | 37.7 |
| 0.5 : 0.5 | 41.3 | 38.0 |
| 0.8 : 0.2 (TDnet) | **41.6** | **38.2** |
| 1.0 : 0.0 | 41.4 | 38.1 |

For both YOLOv10 and YOLOv11, inserting the TDnet MTL head consistently improves $\text{mAP}_{\text{IVT}}$ at IoU 50 and 95, indicating that TDnet provides effective and architecture-agnostic gains.

**Sensitivity to self-distillation fusion weights.** We further study how TDnet performance changes under different fusion weights between the multi-task and self-distillation losses on a single validation fold. Let $\lambda_{\text{MTL}} : \lambda_{\text{SD}}$ denote the relative weights; results are given in Table 11.

The proposed TDnet setting $(0.8 : 0.2)$ achieves the best $\text{mAP}_{\text{IVT}}$, and neighboring configurations perform similarly well, suggesting that the self-distillation fusion is effective and not overly sensitive to moderate weight changes.

**Long-tail Generalization and Rare-triplet Analysis.** We explicitly analyze long-tail generalization on ProstaTD. Appendix F.5 (Per-class Average Precision of IVT Components) reports per-class AP for instruments, verbs, and targets, showing that TDnet improves both frequent and rare classes. To further assess generalization to rare and unseen IVT combinations, we evaluate mAP on triplets with fewer than 300 instances in the training set. Results under five-fold cross-validation are reported in Table 12.

TDnet outperforms YOLOv12 on both global and video-wise metrics for rare triplets, indicating improved generalization to long-tail IVT combinations. Complementing this quantitative analysis, Appendix F.4 provides qualitative case studies. For instance, *thread* is thin and often low-contrast, so the triplet (*grasper*, *grasp*, *thread*) is frequently predicted as (*grasper*, *null*, *null*), illustrating the practical difficulties of rare and visually subtle targets and motivating future research on long-tail modeling in surgical IVT detection.

### F.4  CONFUSION MATRIX ANALYSIS

Based on five-fold cross-validation, Fig. 9 reports the TDnet confusion matrix with columns normalized by ground truth counts, so each column sums to one and values denote recall for each true class. Several common and visually distinctive triplets are comparatively easy to recognize and therefore form a strong diagonal, including (scissors,null,null), (forceps,retract,bladder), and (scissors,cut,bladder).

The off-diagonal structure reveals consistent confusions driven by visual similarity and contextual coupling. Instruments from the same family share geometry and appearance, and single-frame evidence can make actions such as grasp and retract hard to separate. Overall, the majority of confusions occur between triplets involving the same instrument, indicating that most errors arise from misrecognizing the action or target rather than the instrument itself. For example, thread is thin and often low-contrast, which explains why (grasper,grasp,thread) is frequently predicted as (grasper,null,null) with 54.5%. Context can also couple action and target. Suction near the Endobag resembles aspirator near fluid, which leads to (aspirator,retract,Endobag) being predicted

Table 12: Five-fold cross-validation performance on rare triplets (fewer than 300 instances in the dataset). We report IVT-level mAP at IoU 0.5, both globally and video-wise.

| Method | mAP$_{IVT}$ (Global) | mAP$_{IVT}$ (Video-wise) |
|---|---|---|
| YOLOv12 | 19.8 | 20.7 |
| TDnet | **20.6** | **22.1** |

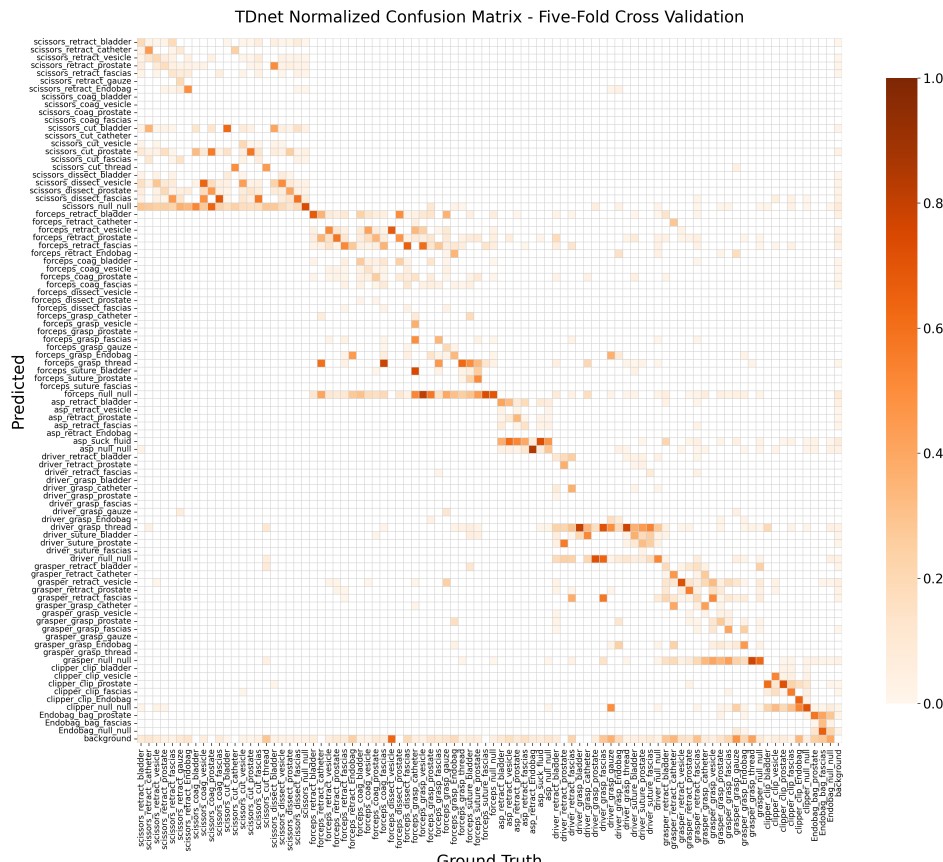

Figure 9: TDnet confusion matrix under five-fold cross-validation on ProstaTD. Columns are normalized by ground truth counts and values show recall for each true class. The x-axis shows ground truth and the y-axis shows predictions. Darker cells indicate higher recall. The diagonal shows correct triplet predictions and the off-diagonal structure reveals within-triplet confusions. Abbreviations in labels are asp for aspirator, driver for driver, clipper for clipper applier, coag for coagulate, and vesicle for vesicle.

as (aspirator,suck,fluid) with 53.8%. In addition, rare triplets with limited examples show frequent mutual confusions such as (driver,grasp,gauze).

In addition, we provide separate confusion matrices for the instrument, action, and target components (Fig. 10). They show that instrument confusions are rare, while actions such as *Coagulate*/*Retract* (42%) and *Suck*/*Grasp* (38%) are frequently confused, and non-organ targets (catheter, gauze, Endobag) are often missed or misclassified as other organs. Consistent with the triplet-level confusion matrix, these component-wise results highlight that the key challenges lie in recognizing actions and targets under long-tailed, visually ambiguous conditions.

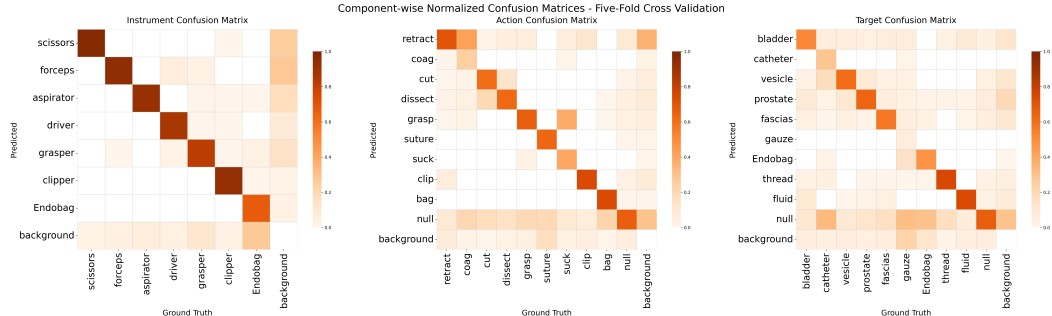

Figure 10: Single component normalized confusion matrices of TDnet under five-fold cross-validation on ProstaTD, for instruments (left), actions (middle), and targets (right). Instrument confusions are generally rare, whereas actions and non-organ targets exhibit strong confusion patterns, indicating that action and target recognition under long-tailed distributions remain the main challenges.

### F.5 PER-CLASS AVERAGE PRECISION OF IVT COMPONENT

To highlight TDnet's contribution to balancing the triplet distribution, we report per-class $AP_{IVT}$ at an IoU threshold of 0.50 under both global and video-wise protocols, computed as five-fold means. The full results are listed in Table 13, and the protocol definitions follow Section D.

Overall, TDnet outperforms the YOLOv12 baseline on the majority of IVT categories for both global and video-wise evaluation. The gains appear not only on frequent and visually distinctive triplets but also on many rare categories, indicating that the combination of multi-task supervision and self distillation improves class balance in practice. While a few classes still favor YOLOv12, the prevailing trend is in favor of TDnet.

These per-class results complement the confusion matrix in Fig. 9. The hardest categories remain those with very low support or subtle action and target cues, which show larger variance. Even so, the per-class improvements align with the aggregate advantages reported in Tables 4 and 8.

Table 13: Comparison of YOLOv12 and TDnet Average Precision (AP) for IVT (Instrument-Action-Target) Classes. Values are five-fold cross-validation means. Higher values are highlighted in bold with light green background. Here, "driver" refers to "needle driver", and "vesicle" abbreviates "seminal vesicle".

| Triplet Class (I-V-T) | YOLOv12 | | TDnet | |
|---|---|---|---|---|
| | Global AP | Video-wise AP | Global AP | Video-wise AP |
| (scissors, retract, bladder) | **23.6** | **21.0** | 21.5 | 18.7 |
| (scissors, retract, catheter) | 56.8 | 61.2 | **71.5** | **70.2** |
| (scissors, retract, vesicle) | **24.7** | 25.7 | 23.5 | **27.9** |
| (scissors, retract, prostate) | 20.2 | 22.4 | **23.7** | **24.1** |
| (scissors, retract, fascias) | **17.9** | 16.8 | 17.6 | **18.7** |
| (scissors, retract, gauze) | **61.4** | **61.4** | 58.3 | 58.3 |
| (scissors, retract, Endobag) | 36.4 | 45.0 | **63.5** | **54.8** |
| (scissors, coagulate, prostate) | **33.0** | **33.0** | 0.0 | 0.0 |
| (scissors, coagulate, fascias) | 0.0 | 0.0 | **45.8** | **45.8** |
| (scissors, cut, bladder) | 64.4 | 69.0 | **65.1** | **74.4** |
| (scissors, cut, vesicle) | 33.2 | 29.7 | **42.2** | **35.8** |
| (scissors, cut, prostate) | 60.1 | 64.1 | **61.2** | **65.5** |
| (scissors, cut, fascias) | 4.8 | **7.0** | **7.3** | 6.8 |
| (scissors, cut, thread) | 54.0 | 45.9 | **57.5** | **50.0** |
| (scissors, dissect, vesicle) | 48.4 | 52.2 | **49.6** | **55.3** |
| (scissors, dissect, prostate) | **42.7** | 38.4 | 42.4 | **40.7** |

Table 13 – continued from previous page

| Triplet Class (I-V-T) | YOLOv12 | | TDnet | |
|---|---|---|---|---|
| | Global AP | Video-wise AP | Global AP | Video-wise AP |
| (scissors, dissect, fascias) | 61.8 | 57.2 | **65.0** | **61.0** |
| (scissors, null, null) | 77.2 | 78.2 | **78.3** | **79.2** |
| (forceps, retract, bladder) | 65.8 | 54.6 | **68.7** | **57.6** |
| (forceps, retract, vesicle) | 53.8 | 56.1 | **56.2** | **60.2** |
| (forceps, retract, prostate) | 60.5 | 60.0 | **62.2** | **60.8** |
| (forceps, retract, fascias) | 54.9 | 41.4 | **57.8** | **45.0** |
| (forceps, coagulate, bladder) | **34.0** | **30.4** | 29.4 | 30.0 |
| (forceps, coagulate, vesicle) | 14.1 | 15.0 | **19.6** | **22.0** |
| (forceps, coagulate, prostate) | 26.1 | 30.7 | **35.3** | **37.8** |
| (forceps, coagulate, fascias) | 9.7 | 10.1 | **15.8** | **14.3** |
| (forceps, dissect, fascias) | 7.9 | 4.1 | **8.9** | **8.4** |
| (forceps, grasp, catheter) | **37.4** | **42.9** | 32.7 | 37.2 |
| (forceps, grasp, prostate) | **51.7** | 25.9 | 35.6 | **26.9** |
| (forceps, grasp, fascias) | **40.9** | **37.9** | 37.0 | 37.0 |
| (forceps, grasp, gauze) | **26.4** | 30.3 | 23.4 | **32.5** |
| (forceps, grasp, Endobag) | 28.1 | 52.8 | **32.1** | **62.7** |
| (forceps, grasp, thread) | 69.6 | 60.3 | **70.2** | **60.6** |
| (forceps, suture, bladder) | **26.8** | **22.9** | 8.8 | 12.7 |
| (forceps, suture, prostate) | **54.8** | **54.8** | 46.2 | 46.9 |
| (forceps, null, null) | 70.8 | 71.4 | **73.6** | **74.3** |
| (aspirator, retract, bladder) | 44.3 | 35.6 | **46.6** | **38.2** |
| (aspirator, retract, vesicle) | **3.4** | **2.3** | **3.4** | 2.0 |
| (aspirator, retract, prostate) | **46.6** | 25.5 | 46.5 | **25.9** |
| (aspirator, retract, fascias) | **23.1** | **25.5** | 19.4 | 21.0 |
| (aspirator, suck, fluid) | 63.1 | 62.8 | **64.0** | **64.5** |
| (aspirator, null, null) | 27.7 | 31.9 | **30.9** | **35.9** |
| (driver, retract, bladder) | **23.4** | **25.9** | 14.2 | 19.4 |
| (driver, retract, prostate) | 54.6 | 55.3 | **66.8** | **64.2** |
| (driver, grasp, Endobag) | 12.8 | 12.8 | **27.5** | **28.4** |
| (driver, grasp, thread) | 80.0 | 76.9 | **80.6** | **77.3** |
| (driver, suture, bladder) | 40.6 | 44.8 | **42.5** | **45.9** |
| (driver, suture, prostate) | **46.7** | 53.3 | 40.9 | **51.7** |
| (driver, null, null) | 60.8 | 63.8 | **61.4** | **64.5** |
| (grasper, retract, bladder) | 22.1 | 18.9 | **34.9** | **24.1** |
| (grasper, retract, catheter) | **45.6** | 69.2 | 36.5 | **73.8** |
| (grasper, retract, vesicle) | **78.4** | **73.9** | 74.1 | 72.8 |
| (grasper, retract, prostate) | 59.6 | 54.1 | **60.5** | **58.7** |
| (grasper, retract, fascias) | **26.1** | **24.0** | 17.5 | 17.9 |
| (grasper, grasp, catheter) | 57.1 | **71.7** | **59.5** | 67.7 |
| (grasper, grasp, prostate) | **45.9** | **52.6** | 37.2 | 46.2 |
| (grasper, grasp, fascias) | **51.6** | **54.7** | 51.5 | 47.3 |
| (grasper, grasp, Endobag) | 43.3 | 42.0 | **55.8** | **65.2** |
| (grasper, null, null) | **67.4** | 62.1 | 66.4 | **62.7** |
| (clip applier, clip, vesicle) | 63.0 | 55.1 | **66.3** | **61.8** |
| (clip applier, clip, prostate) | 59.9 | 72.8 | **65.2** | **75.7** |
| (clip applier, clip, fascias) | 56.4 | 51.8 | **66.5** | **58.9** |
| (clip applier, clip, Endobag) | **54.6** | **53.9** | 50.6 | 49.7 |
| (clip applier, null, null) | 72.8 | 70.6 | **76.7** | **75.3** |
| (Endobag, bag, prostate) | 47.9 | 65.5 | **49.9** | **68.2** |
| (Endobag, bag, fascias) | 28.3 | 28.3 | **30.8** | **30.8** |
| (Endobag, null, null) | 21.8 | 22.9 | **24.7** | **35.5** |

### F.6 Cost-Aware Ordinal IVT Evaluation

To move beyond a binary evaluation that counts an entire IVT prediction as wrong as soon as any single component is incorrect, we design an ordinal, cost-aware metric that reflects the different importance of each component (bbox, instrument, action, and target), rather than treating all errors equally. In particular, we encode the ordinal preference "instrument > action ≈ target" directly in the scoring rule, and we follow the standard detection protocol to associate predictions with ground-truth boxes: predictions are sorted by confidence and greedily matched to at most one ground-truth box each (per image), with each ground truth used at most once, similar to mAP evaluation. *Note that this represents only a preliminary proposal for a cost-aware ordinal IVT evaluation metric, and some implementation details may still require further refinement and justification.*

**Per-prediction cost.** All formulas below are first defined for a single IVT class. We perform greedy IoU-based matching with threshold $0.5$. For each prediction $p$, if it cannot be matched to any ground-truth box with $\mathrm{IoU} \geq 0.5$, its cost-aware score is $0$; otherwise, we assign a non-zero score:

$$\text{score}_{\text{IVT}}(p) = \begin{cases} 0, & \text{if no matched GT with IoU} \geq 0.5, \\ s_{\text{ivt}}(p), & \text{otherwise.} \end{cases} \tag{11}$$

For matched pairs, we assign partial credit to each component:

$$s_{\text{ivt}}(p) = 0.1 + 0.5 \cdot \mathbb{K}[\text{instrument correct}] + 0.2 \cdot \mathbb{K}[\text{action correct}] + 0.2 \cdot \mathbb{K}[\text{target correct}], \tag{12}$$

where $\mathbb{K}[\cdot]$ is an indicator for correct prediction of the corresponding component. The constant $0.1$ reflects that the bbox is already correct under the $\mathrm{IoU} \geq 0.5$ condition, and thus indicates that at least "something" is happening in that region, even if the instrument, action, or target labels are not fully correct. The weight $0.5$ encodes the higher ordinal importance of the instrument, while actions and targets share a smaller but equal weight ($0.2$ each), and the weights sum to $1.0$.

**Cost-aware precision, recall, and F1.** Let $\mathcal{P}$ and $\mathcal{G}$ be the sets of predictions and ground truths for this single IVT class. Each prediction $p \in \mathcal{P}$ is matched to at most one ground truth, and each $g \in \mathcal{G}$ is matched to at most one prediction. Unmatched predictions and unmatched ground truths receive zero score.

We define the cost-aware precision and recall as simple averages of the per-prediction scores:

$$P_{\text{cost}} = \frac{1}{|\mathcal{P}|} \sum_{p \in \mathcal{P}} \text{score}_{\text{IVT}}(p), \tag{13}$$

$$R_{\text{cost}} = \frac{1}{|\mathcal{G}|} \sum_{g \in \mathcal{G}} \text{score}_{\text{IVT}}(g), \tag{14}$$

where $\text{score}_{\text{IVT}}(g)$ is the score of the prediction matched to $g$ (or $0$ if $g$ is unmatched). We then combine them in a cost-aware F1-style score:

$$F1_{\text{cost}} = \frac{2 \cdot P_{\text{cost}} \cdot R_{\text{cost}}}{P_{\text{cost}} + R_{\text{cost}} + \varepsilon}, \tag{15}$$

with a small $\varepsilon$ to avoid division by zero.

**Single-component scores.** Using the same matching for this class, we define separate scores for each component. Unmatched predictions (and unmatched GTs) always contribute $0$ for all components. For a matched prediction $p$:

$$\text{score}_{\text{Bbox}}(p) = 0.1, \tag{16}$$
$$\text{score}_{\text{I}}(p) = \mathbb{K}[\text{instrument correct}], \tag{17}$$
$$\text{score}_{\text{V}}(p) = \mathbb{K}[\text{action correct}], \tag{18}$$
$$\text{score}_{\text{T}}(p) = \mathbb{K}[\text{target correct}], \tag{19}$$

and $\text{score}_c(p) = 0$ for all $c \in \{\text{Bbox}, \text{I}, \text{V}, \text{T}\}$ if $p$ is unmatched. Plugging $\text{score}_c$ into Eqs. equation 13–equation 15 (instead of $\text{score}_{\text{IVT}}$) yields per-class $P_{\text{cost}}^c$, $R_{\text{cost}}^c$, and $F1_{\text{cost}}^c$ for bbox, instrument, action, and target.

Table 14: Cost-aware F1 scores (5-fold class-wise macro average over IVT classes) for bbox, instrument, action, target, and overall IVT. The overall IVT score is bounded by $1.0$, with a maximum of $0.1$ from bbox IoU matching and $0.5/0.2/0.2$ from correct instrument/action/target labels, respectively.

| Method | $F1_{\text{cost}}^{\text{Bbox}}$ | $F1_{\text{cost}}^{\text{I}}$ | $F1_{\text{cost}}^{\text{V}}$ | $F1_{\text{cost}}^{\text{T}}$ | $F1_{\text{cost}}^{\text{IVT}}$ |
|---|---|---|---|---|---|
| RT-DETR | 0.0491 | 0.2435 | 0.0859 | 0.0876 | 0.4635 |
| YOLOv12 | 0.0503 | 0.2508 | 0.0856 | 0.0836 | 0.4676 |
| TDnet | 0.0527 | 0.2627 | 0.0895 | 0.0880 | 0.4871 |

Table 15: Leave-one-source-out IVT performance across hospitals on ProstaTD. For each row, models are trained on the remaining two sources and evaluated on the held-out source. Metrics are IVT-level mAP$_{50}$, mAP$_{95}$, and F1 (in %).

| Leave-out source | RT-DETR | | | YOLOv12 | | | TDnet | | |
|---|---|---|---|---|---|---|---|---|---|
| | mAP$_{\text{IVT,50}}$ | mAP$_{\text{IVT,95}}$ | F1 | mAP$_{\text{IVT,50}}$ | mAP$_{\text{IVT,95}}$ | F1 | mAP$_{\text{IVT,50}}$ | mAP$_{\text{IVT,95}}$ | F1 |
| ESAD | 19.3 | 16.4 | 16.8 | 18.4 | 16.3 | 17.5 | 21.1 | 18.5 | 20.3 |
| PSI-AVA | 22.1 | 20.1 | 19.9 | 20.6 | 18.7 | 20.7 | 23.1 | 20.6 | 23.4 |
| PWH | 15.4 | 13.2 | 13.9 | 16.0 | 14.3 | 15.5 | 16.2 | 14.4 | 16.0 |
| Average | 18.9 | 16.6 | 16.9 | 18.3 | 16.4 | 17.9 | 20.1 | 17.8 | 19.9 |

**Aggregation over IVT classes.** The above definitions are all per-class. Let $k \in \{1, \ldots, K\}$ index IVT classes, and let $F1_{\text{cost}}^{(k)}$ and $F1_{\text{cost}}^{(k,c)}$ denote the overall and single-component F1 scores for class $k$ computed by Eqs. equation 15 and equation 16–equation 19. Our final reported metrics are the macro-averages over classes:

$$\overline{F1}_{\text{cost}} = \frac{1}{K}\sum_{k=1}^{K} F1_{\text{cost}}^{(k)}, \qquad \overline{F1}_{\text{cost}}^{c} = \frac{1}{K}\sum_{k=1}^{K} F1_{\text{cost}}^{(k,c)}, \tag{20}$$

which we report as the cost-aware F1 for overall IVT and for each single component (bbox, instrument, action, target).

This cost-aware metric is used as a complementary analysis tool to standard mAP/mAR, providing an ordinal and decomposable view of how much error comes from instruments, actions, and targets.

From Table 14, we observe a qualitatively different behavior compared to the single-component mAP metrics. While single-component mAP is computed instrument-wise, action-wise, or target-wise, our new cost-aware metric is defined IVT-class-wise and thus penalizes all components jointly at the triplet level. As a result, even bounding boxes, instruments, actions, and targets all show noticeable room for improvement, especially for rare triplets where every component tends to perform poorly. Nevertheless, our proposed TDnet consistently achieves the best, indicating that its advantage remains under this stricter, triplet-aware evaluation.

As for more complex ordinal schemes, for example prioritizing certain actions or targets more highly in specific clinical scenarios, this would require detailed input from expert surgeons and a task-specific cost design. We view such class- or scenario-specific ordinal cost modeling as promising future work beyond the scope of the current paper.

F.7 LEAVE-ONE-SOURCE-OUT EVALUATION

To assess cross-hospital domain generalization, we conduct leave-one-source-out experiments on ProstaTD, where models are trained on two hospitals and evaluated on the held-out one. We report IVT-level mAP$_{50}$, mAP$_{5095}$, and F1 score for RT-DETR, YOLOv12, and TDnet across all permutations in Table 15.

Compared with the intra-hospital setting, all methods suffer a substantial drop in IVT detection performance in the leave-one-source-out configuration, indicating that cross-hospital IVT generalization is challenging on ProstaTD and highlighting its value as a benchmark for future domain generalization and continual learning methods.

## G LIMITATIONS AND FUTURE WORKS

### G.1 LONG-TAIL DISTRIBUTION AND RARE TRIPLET CASES

Similar to CholecT50 (Nwoye et al., 2022), our ProstaTD dataset also exhibits uneven data distribution, with a significant proportion of rare triplets. These uncommon instances often arise from situational constraints, such as the time cost of instrument switching, where surgeons temporarily repurpose available instruments to perform alternative actions. In addition, some triplets may be missing from the dataset due to anatomical differences between patients, which can lead to additional dissection steps or unconventional surgical techniques. In real-world procedures, the potential number of triplet combinations is far greater, influenced by variations in surgical habits, patient conditions, and even occasional errors. However, these rare and abnormal cases are difficult to capture due to their infrequent occurrence. Despite their low frequency, they are highly relevant to clinical safety and robustness, making their inclusion vital in medical datasets. Moreover, most current approaches constrain triplet recognition to a fixed, predefined class set, which inherently prevents the identification of unseen or abnormal triplets. This limitation underscores the need for models that independently predict instruments, actions, and targets, and then flexibly compose triplet candidates, rather than relying on rigid class enumeration. Although our TDnet is specifically designed for triplet class imbalance, it is clear that there is still significant room for improvement on this problem. Therefore, our ProstaTD dataset provides a venue to address this challenge, facilitating future research in surgical action recognition.

### G.2 TEMPORAL MODELING

Using temporal information for the surgical triplet tasks is nontrivial. For the surgical triplet **classification** task, RiT (Sharma et al., 2023) reports a clear gain on the verb metric $mAP_V$ from 62.0% to 64.0%, but only a small gain on the full triplet metric $mAP_{IVT}$ from 29.4% to 29.7%, indicating that in their method temporal cues help verbs more than the complete triplet. In our surgical triplet **detection** setting, we also evaluated a temporal multi-task model in the style of TAPIR (Valderrama et al., 2022) and observed that, under our setup, the results in Table 4 are limited and in some cases worse than the single-frame baseline. Since such temporal modules add computation and latency, this cost becomes nontrivial for real-time or resource-constrained deployment and does not offset the limited accuracy gains we observed. Modern pipelines commonly perform detection per frame, while temporal consistency is handled in a downstream tracking stage through identity association and tracking supervision. For instance, the recent SurgiTrack (Nwoye & Padoy, 2025) adopts a two-stage approach to achieve fine-grained multi-class multi-instrument tracking. The first stage focuses on using state-of-the-art per-frame detection models for initial localization. Subsequently, the tracking stage leverages temporal consistency across frames to maintain object association and provide more precise dynamic trajectories for minimally invasive surgeries. Guided by these observations, we do not add temporal modules to TDnet at the detection stage and instead plan to exploit temporal information in a future extension that integrates tracking, where identity supervision can utilize temporal continuity more effectively.

### G.3 CONTINUAL AND CROSS-DOMAIN LEARNING

Our benchmark currently consists of three datasets from different sources, which naturally differ in visual appearance, recording settings, and annotation distributions. The experiments in Appendix F.7 suggest that models trained on one source do not trivially generalize to the others, indicating substantial room for improvement in continual learning and cross-domain generalization within this surgical domain.

At present, TDnet is designed as a single-domain detector and does not explicitly incorporate mechanisms for domain adaptation, domain generalization, or continual learning across datasets (e.g., avoiding catastrophic forgetting when adapting from one hospital to another). Designing architectures and training strategies that can leverage all three sources jointly, or adapt sequentially without performance degradation, is therefore an important direction for future work that ProstaTD and the associated datasets can help to systematically evaluate.

### G.4 Bridging to Robotic Assistance and Simulation

Our current work focuses on perception, and the role of robotic assistance is only partially explored. Spatially grounded triplet detection is particularly relevant for robotic assistance tasks such as aspiration control, where systems must reason about *where* interactions occur to ensure safe movement, avoid collisions, or trigger alerts. Labels alone are insufficient, and bounding boxes provide the necessary spatial context for action execution, navigation, and enforcing no-go zones.

Recent work such as SRT-H (Kim et al., 2025) has already demonstrated that robots can autonomously perform multi-step surgical procedures with minimal human intervention. In such systems, the robot must operate on structured representations of "what tool is doing what to which anatomy" at each moment. Our ProstaTD dataset and triplet detection task provide exactly this kind of structured, spatially grounded information in the form of *instrument–action–target* triplets.

Although we do not yet integrate our models into closed-loop robotic systems or simulators, a promising future direction is to use ProstaTD to supply these triplet-based states for training and evaluating autonomous surgical policies. For example, a robot could adapt its aspiration behavior or enforce safety constraints directly based on detected instrument–action–target triplets around critical structures. Realizing this perception–control integration lies beyond the scope of the present work but is a key long-term goal that our dataset is designed to support.

## H Supplementary Information on Annotation Software

### H.1 Imported JSON File Example

In this part, we provide additional details about the imported JSON file format, complementing the description in Appendix C. As mentioned earlier, both annotation tools, Triplet-labelme and SurgLabel, rely on this JSON file as input. Once imported, the software automatically parses the file and determines whether an attribute corresponds to *image-level* or *instance-level* annotation based on the presence of the keyword "image". In our surgical triplet detection task, only *instance-level* annotations are used.

The example below shows the schema used for our surgical triplet detection task. In addition, we provide three customizable examples to illustrate how users can extend the schema to meet different needs, such as adding spatial position information, switching to image-level annotation, or combining image-level and instance-level attributes. After import, the relevant attributes specified in the JSON file are displayed in the right-hand panel of the annotation interface, as shown in Fig. 6 and Fig. 7. This adaptive design allows our tools to support not only surgical annotation but also diverse annotation tasks across other domains.

**Imported JSON Label Structures for Our Annotation Software**

```json
{
  "tools": {
    "scissors": {
      "actions": ["retract", "coagulate", "cut", "dissect", "null"],
      "targets": ["bladder", "catheter", "seminal vesicle", "prostate
", "fascias", "gauze", "Endobag", "thread", "null"]
    },
    "forceps": {
      "actions": ["coagulate", "retract", "grasp", "dissect", "bag",
"suture", "null"],
      "targets": ["bladder", "catheter", "seminal vesicle", "prostate
", "fascias", "gauze", "thread", "Endobag", "null"]
    },
    "aspirator": {
      "actions": ["retract", "suck", "null"],
      "targets": ["bladder", "catheter", "seminal vesicle", "prostate
", "fascias", "fluid", "Endobag", "null"]
    },
    "needle driver": {
```

```
      "actions": ["retract", "grasp", "dissect", "suture", "null"],
      "targets": ["bladder", "catheter", "seminal vesicle", "prostate
    ", "fascias", "gauze", "Endobag", "thread", "null"]
    },
    "grasper": {
      "actions": ["retract", "grasp", "dissect", "null"],
      "targets": ["bladder", "catheter", "seminal vesicle", "prostate
    ", "fascias", "gauze", "Endobag", "thread", "null"]
    },
    "clip applier": {
      "actions": ["clip", "null"],
      "targets": ["bladder", "catheter", "seminal vesicle", "prostate
    ", "fascias", "Endobag", "null"]
    },
    "Endobag": {
      "actions": ["bag", "null"],
      "targets": ["bladder", "catheter", "seminal vesicle", "prostate
    ", "fascias", "null"]
    }
}

// --------------------
// Custom Example 1 add spatial position annotation
// "tools": {
//   "scissors": {
//     "actions": [...],
//     "targets": [...],
//     "position": ["left", "right", "anterior", "posterior", "null
  "]
//   }
// }

// --------------------
// Custom Example 2 switch to image level annotation
// "image": {
//   "phase": ["suturing", "clipping", "dissection", ...]
// }

// --------------------
// Custom Example 3 combine image-level and instrument-level
  annotation
// {
//   "image": {
//     "phase": ["suturing", "clipping"],
//     "lighting": ["none", "minor", "severe", ...],
//     "smoke": ["yes", "no"],
//     ...
//   },
//   "tools": {
//     "forceps": {
//       "actions": [...],
//       "targets": [...],
//       "position": [...],
//       ...
//     }
//   "targets": {
//     "DVC": {
//       "bleeding": ["yes", "no"]
//       ...
//       }
//     }
//   }
// }
```

```
    // Note it is not required to restrict the schema to "image" or "
      tools"
    // Users can design their own JSON structure depending on the
      annotation task
    // even in the case of natural images or other domains
    // Our annotation tool can adaptively handle such structures
    // enabling modifications on a single frame or across the temporal
      dimension
  }
```

## H.2 CUSTOMIZABLE USER INTERFACE FEATURES

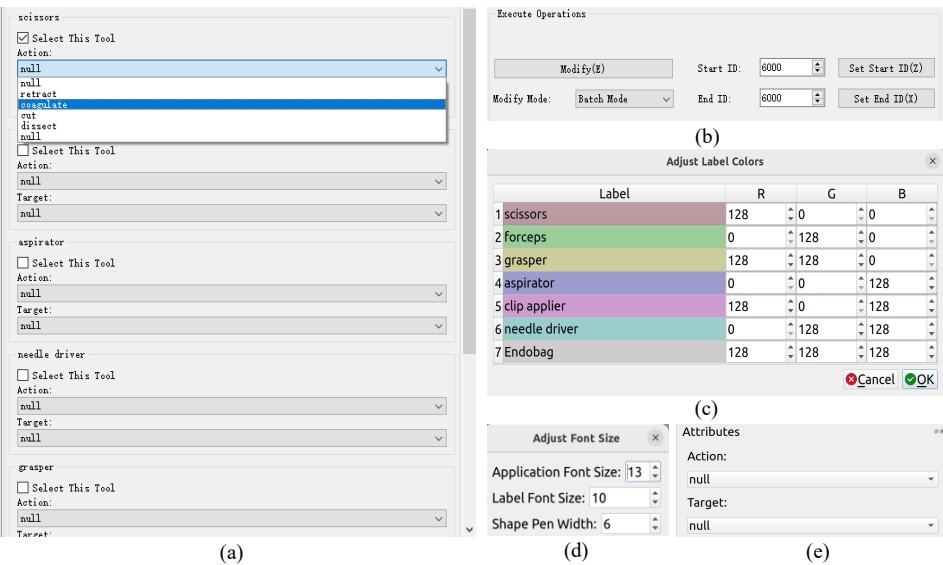

Figure 11: **Customizable user interface features in our annotation tools.** (a) SurgLabel automatically generates instrument selection panels based on the imported JSON schema. (b) Temporal annotation interface in SurgLabel after tool selection. (c, d) Examples of customizable options in Triplet-labelme and SurgLabel after JSON import, including adjustable colors, font weights, and other visual settings. (e) Instrument selection panel automatically generated in Triplet-labelme according to the JSON schema.

After importing the JSON file, both Triplet-labelme (Fig. 6) and SurgLabel (Fig. 7) automatically parse the schema and generate instrument selection panels, as illustrated in Fig. 11(a) and (e). In Triplet-labelme, users can select a specific instrument instance and then perform instance-level modifications of the corresponding action and target in a single frame. In contrast, SurgLabel supports batch modifications through the temporal annotation interface shown in Fig. 11(b). Furthermore, both tools provide customizable options, such as adjusting bounding box thickness and color, label font size, or transparency, as demonstrated in Fig. 11(c) and (d). These flexible user interface features facilitate efficient annotation and allow adaptation to diverse user preferences and annotation requirements.

## I THE USE OF LARGE LANGUAGE MODELS

In the preparation of this manuscript, a Large Language Model (LLM) was used solely as a writing aid for grammar checking and refining unclear expressions, ensuring clarity and coherence in the presentation of our work.

APPENDIX REFERENCES

Vivek Singh Bawa, Gurkirt Singh, Francis KapingA, Inna Skarga-Bandurova, Elettra Oleari, Alice Leporini, Carmela Landolfo, Pengfei Zhao, Xi Xiang, Gongning Luo, et al. The saras endoscopic surgeon action detection (esad) dataset: Challenges and methods. *arXiv preprint arXiv:2104.03178*, 2021.

Shuangchun Gui, Zhenkun Wang, Jixiang Chen, Xun Zhou, Chen Zhang, and Yi Cao. Mt4mtl-kd: A multi-teacher knowledge distillation framework for triplet recognition. *IEEE Transactions on Medical Imaging*, 43(4):1628–1639, 2023.

Ji Woong Kim, Juo-Tung Chen, Pascal Hansen, Lucy Xiaoyang Shi, Antony Goldenberg, Samuel Schmidgall, Paul Maria Scheikl, Anton Deguet, Brandon M White, De Ru Tsai, et al. Srt-h: A hierarchical framework for autonomous surgery via language-conditioned imitation learning. *Science robotics*, 10(104):eadt5254, 2025.

Chinedu Innocent Nwoye and Nicolas Padoy. Data splits and metrics for benchmarking methods on surgical action triplet datasets. *arXiv preprint arXiv:2204.05235*, 2022.

Chinedu Innocent Nwoye and Nicolas Padoy. Surgitrack: Fine-grained multi-class multi-tool tracking in surgical videos. *Medical Image Analysis*, 101:103438, 2025.

Chinedu Innocent Nwoye, Tong Yu, Cristians Gonzalez, Barbara Seeliger, Pietro Mascagni, Didier Mutter, Jacques Marescaux, and Nicolas Padoy. Rendezvous: Attention mechanisms for the recognition of surgical action triplets in endoscopic videos. *Medical Image Analysis*, 78:102433, 2022.

Saurav Sharma, Chinedu Innocent Nwoye, Didier Mutter, and Nicolas Padoy. Rendezvous in time: an attention-based temporal fusion approach for surgical triplet recognition. *International Journal of Computer Assisted Radiology and Surgery*, 18(6):1053–1059, 2023.

Natalia Valderrama, Paola Ruiz Puentes, Isabela Hernández, Nicolás Ayobi, Mathilde Verlyck, Jessica Santander, Juan Caicedo, Nicolás Fernández, and Pablo Arbeláez. Towards holistic surgical scene understanding. In *International conference on medical image computing and computer-assisted intervention*, pp. 442–452. Springer, 2022.

Amine Yamlahi, Thuy Nuong Tran, Patrick Godau, Melanie Schellenberg, Dominik Michael, Finn-Henri Smidt, Jan-Hinrich Nölke, Tim J Adler, Minu Dietlinde Tizabi, Chinedu Innocent Nwoye, et al. Self-distillation for surgical action recognition. In *International Conference on Medical Image Computing and Computer-Assisted Intervention*, pp. 637–646. Springer, 2023.

Yifu Zhang, Peize Sun, Yi Jiang, Dongdong Yu, Fucheng Weng, Zehuan Yuan, Ping Luo, Wenyu Liu, and Xinggang Wang. Bytetrack: Multi-object tracking by associating every detection box. In *European conference on computer vision*, pp. 1–21. Springer, 2022.

Yian Zhao, Wenyu Lv, Shangliang Xu, Jinman Wei, Guanzhong Wang, Qingqing Dang, Yi Liu, and Jie Chen. Detrs beat yolos on real-time object detection. In *Proceedings of the IEEE/CVF conference on computer vision and pattern recognition*, pp. 16965–16974, 2024.

