# OpenReview forum: "ProstaTD: Bridging Surgical Triplet from Classification to Fully Supervised Detection"
_ICLR.cc/2026/Conference — ICLR 2026 Poster_

### Official Review · Reviewer_PPXn · 2025-10-31

**Soundness:** 4
**Presentation:** 3
**Contribution:** 4
**Rating:** 8
**Confidence:** 3

**Summary:**

ProstaTD is a large-scale dataset for surgical triplet detection in robot-assisted prostatectomy videos. It contains 71,775 annotated frames with 196,490 triplet instances from 21 surgeries across three institutional sources. The dataset addresses limitations in existing surgical video datasets by providing precise spatial bounding box annotations rather than relying solely on image-level labels, and by implementing clinically grounded temporal boundaries for surgical actions. The annotation taxonomy consists of 7 instrument types, 10 actions, and 10 anatomical targets. All annotations were created under medical supervision to ensure clinical validity. The temporal labelling scheme distinguishes between continuous actions (such as grasping) and momentary actions (such as cutting), which improves annotation consistency compared to prior datasets. The dataset exhibits higher surgical complexity than comparable laparoscopic datasets, with more concurrent instruments per frame reflecting actual surgical conditions. The authors benchmarked multiple state-of-the-art detection architectures on ProstaTD. Results show that fully supervised models using bounding box annotations substantially outperform weakly supervised methods that use only class labels. The authors propose TDnet, a new architecture that combines spatial detection with action-target recognition.

**Strengths:**

The paper uses a cross institutional dataset to create a dataset with detailed surgical annotations supervised by experts. Usually the available datasets in this area come from a single institution and one or few surgeons, and the limited variability is known to affect the results. The additional variability will allow testing the generalisation of these methods more thoroughly

The dataset will be open sourced and baselines of appropriate methods are provided for the predictive Task. The authors also proposed their own variation of a Neural Network that outperforms commonly used methods in the field.

The dataset includes bounding boxes of the triplets, which is not offered by previous datasets, and it contains more labeled instances of triplets (tool, action, anatomical object) than any previous dataset. The earlier datasets in the field that offer bounding boxes of triplet instances have less than half the annotated instances than this dataset.

**Weaknesses:**

The details of the proposed Neural Network, TDnet, are not in the main paper but in the appendix. Although the main contribution is an open source dataset, ICLR is mostly a Machine Learning conference and putting forward an algorithmic proposition for the proposed task would make.

While TDnet is sensible, including new dependencies in the labels to the optimisation process, it would also be a lot more interesting to propose substantially more innovative algorithmic propositions. Also, TDnet as the best performing model is significant enough to be presented in the main paper and not the appendix.

I am not sure what is the role bounding box detection in this problem. Usually you need a bounding box to identify interactions between objects in an image or something of that sort. However, if you have triplets that consider action you already cover that with a classification problem. I assume the authors have some aspiration in robotics that was not clarified in the paper.

**Questions:**

How do you plan to address the class imbalance of rare triplets ?

In which clinical context does object detection offer information that the triplet (with action) classification does not account for?

**Details Of Ethics Concerns:**

The dataset contains surgical videos on humans.
It seems to be anonymised but in such tasks indirect identification is sometimes possible.

---

> ### Author Response · Authors · 2025-11-20
> **Response to Reviewer PPXn (Part 1)**
>
> ### W1. "Details of proposed TDnet not in main paper but in appendix."
> Thank you for the suggestion; we have moved the key components of TDnet from the appendix into the main text.
>
> ### W2. "it would also be a lot more interesting to propose substantially more innovative algorithmic propositions."
>
> We thank the reviewer for this comment. This work is primarily a dataset and benchmark paper: the main contribution is the ProstaTD dataset together with a reproducible annotation and evaluation protocol, in line with prior dataset‑centric works such as Cholectrack20 [1] and SPORTU [2]. This provides a solid basis for future method development and fair comparison.
>
> TDnet is introduced mainly as a practical new baseline to characterize the difficulty of ProstaTD and to offer a clear, shared reference point. We agree that there is ample room for more innovative architectures, and we expect ProstaTD to serve as the platform on which such methods can be developed and objectively evaluated in future work. In this sense, we see the main value of the current paper as establishing the common ground on which more ambitious algorithmic advances can be reliably built and compared.
>
> [1] Nwoye, C. I., Elgohary, K., Srinivas, A., Zaid, F., Lavanchy, J. L., & Padoy, N. (2025). Cholectrack20: A multi-perspective tracking dataset for surgical tools. CVPR 2025.
>
> [2] Xia, H., Yang, Z., Zou, J., Tracy, R., Wang, Y., Lu, C., ... & Chen, H. (2025). SPORTU: A Comprehensive Sports Understanding Benchmark for Multimodal Large Language Models. ICLR 2025.
>
> ### W3. "Role of bounding box detection unclear; usually need bounding box to identify interactions between objects, but triplets that consider action cover that with classification problem; aspiration in robotics not clarified in paper."
>
> In our design, bounding box detection and triplet classification serve complementary roles. Detection localizes triplets frame-by-frame, while triplets describe interactions grounded in those detections (e.g., scissors–cut–prostate). A pure triplet classification setup without bounding boxes can only indicate that a given instrument class (e.g., a grasper) is present somewhere in the frame, but cannot resolve which specific instance is acting when multiple identical instruments are visible at the same time. In contrast, instance-level bounding boxes not only disambiguate these cases but also supply spatial context that can be exploited to improve the accuracy and robustness of triplet classification.
>
> Moreover, this spatial grounding is also crucial for robotic assistance tasks such as instrument control, where systems must reason about where interactions occur to ensure safe movement, avoid collisions, or trigger alerts. Recent work such as SRT-H[3] has shown that robots can autonomously execute multi-step surgical procedures, and such systems must effectively understand which instrument is doing what to which anatomy at each moment, which matches the structured information provided by our instrument–action–target triplets. Labels alone are insufficient, and bounding boxes provide the necessary spatial context for action execution, navigation, and enforcing no-go zones.
>
> We have clarified this envisioned role for aspiration-related applications more explicitly in our future work section.
>
> [3] Kim, J. W., Chen, J. T., Hansen, P., Shi, L. X., Goldenberg, A., Schmidgall, S., ... & Krieger, A. (2025). SRT-H: A hierarchical framework for autonomous surgery via language-conditioned imitation learning. Science robotics, 10(104).
>
> ### Q1. "How do you plan to address the class imbalance of rare triplets?"
> Thank you for raising this important point. To address rare triplets, we are expanding the dataset with surgeries involving other organs, which will increase the frequency of currently rare but clinically relevant combinations. Technically, we see ProstaTD as a testbed for methods targeting rare or unseen triplets. Vision–language models (VLMs), combined with scene-graph modeling and long-tail strategies (e.g., re-weighting, specialized losses), offer promising directions for generalizing to semantically plausible but rare interactions. This is especially valuable for modeling underexplored yet clinically important triplets, this offers a path to progressively improve performance on rare, yet clinically important, triplets.

---

> ### Author Response · Authors · 2025-11-20
> **Response to Reviewer PPXn (Part 2)**
>
> ### Q2. "In which clinical context does object detection offer information that the triplet (with action) classification does not account for?"
>
> Thank you for the question. Triplet classification mainly answers what interaction is occurring, but without detection it lacks explicit information about how many objects are involved and which instance of an object participates.
>
> In real surgery, it is very common to have multiple identical instruments in the field. For example, there may be two graspers present at the same time. With triplet classification alone, the system only knows that “grasper–retract–liver” is happening, but it cannot tell whether one or two graspers are present, nor which specific grasper is performing the action. Bounding box detection assigns a separate spatial identity to each instance, allowing the model to reason about multiple tools of the same type, track each one over time, and associate actions with the correct instance. This instance‑level distinction is important for understanding coordination between instruments, which is common in clinical practice.
>
> Moreover, this spatial grounding can in turn improve recognition itself. Knowing where each instrument instance is, and how it moves relative to the scene, helps disambiguate actions and reduce confusion between similar triplets. In this sense, detection and triplet classification are complementary: detection provides instance‑level spatial information, while triplets provide semantic interactions on top of these detections.

---

### Official Review · Reviewer_wQHi · 2025-10-31

**Soundness:** 3
**Presentation:** 3
**Contribution:** 3
**Rating:** 6
**Confidence:** 5

**Summary:**

The paper presents a novel dataset comprising multi-rater triple annotations and instrument bounding boxes for 21 robot-assisted prostatectomies, totaling 71,000 frames, sourced from multiple centers. Moreover, the paper presents the performance of a novel neural network “TDnet” that has been trained on the dataset. The paper also makes the two annotation tools that were used during the dataset's creation available to the public. The appendix, which is extensive in nature, provides insights into a number of areas. These include the annotation software, the model architecture, the experimental setup and the ablation studies.

**Strengths:**

1. important topic that proposes an extensive multi-center multi-rater dataset driving forward the field of triplet detection, encompassing 89 triplets of 7 instruments, 10 actions and 10 targets on 71k frames
2. The dataset is compared to other triplet datasets, clearly pointing out its benefits.
3. Further the paper implements a benchmark using state-of-the-art network architectures and the novel TDnet.

**Weaknesses:**

1. The paper devotes substantial space to criticizing CholecT50 rather than focusing on the independent contributions of the presented dataset.
2. The description of the TDnet architecture, the experimental design and ablation study are not contained in the paper but only in the appendix.

Minor Comments:
M1. The highlighting in Appendix Table 10 is inconsistent, as the higher value is not always highlighted in bold.
M2. The paper cites 7 arXiv sources. Are peer-reviewed publications for these papers available?

**Questions:**

1. The paper claims no bounding boxes are available in CholecT50. However, the datasets overlaps partially with the CholecTrack20 datasets, which provides bounding boxes [1].
Is the annotation available from both datasets comparable to the presented annotations?
2. The paper claims inconsistent annotation boundaries in CholecT50, but at the same time relaxes the temporal definition on “momentary action” to “include up to 2 seconds before and after [] the contact“. How is a consistent temporal annotation between annotators ensured?
How is the start and end defined when it is not the exact time of contact?
3. The dataset merged the triplets of the aspirator sucking fluid, smoke and blood, resulting in the second most common triplet. Especially concerning the intraoperative importance of bleeding, why were these triplets merged?
4. The paper claims a higher complexity compared to CholecT50 as a single frame on average contains more triplets. This is due to the fact that CholecT50 contains laparoscopic surgeries and ProstaTD robot-assisted surgeries. In the later one of the three attached tools is not being used. This is further reflected in the fact that over one quarter of all triplets are made up by an instrument with null action and null target. Is the complexity of the dataset still higher when these null triplets are factored out?
5. The benchmark is evaluated using mAP, Precision, Recall and F1-Score. All these metrics only distinguish correct and incorrect triplets. Does the impact of confusion differ between detecting the wrong tool compared only confusing the action? Could an ordinal metric like estimated cost provide further insights into the actual confusion of instruments, actions, or targets?

[1] https://github.com/CAMMA-public/cholectrack20

---

> ### Author Response · Authors · 2025-11-20
> **Response to Reviewer wQHi (Part 1)**
>
> ### W1. "Paper devotes substantial space to criticizing CholecT50 rather than focusing on independent contributions of presented dataset."
>
> We sincerely respect CholecT50 as a foundational effort in surgical analysis. Our discussion aimed not to criticize it, but to clarify remaining gaps and to motivate the need for a complementary, fully supervised, multi-institutional dataset like ProstaTD.
>
> ### W2. "Description of TDnet architecture, experimental design and ablation study not contained in paper but only in appendix."
>
> Thank you for the comment. As this work is a dedicated dataset contribution, most of our effort has been focused on dataset design and analysis, similar to prior dataset papers[1]. As suggested, we have moved the key TDnet architecture and design into the main text in the revised version.
>
> [1] Nwoye, C. I., Elgohary, K., Srinivas, A., Zaid, F., Lavanchy, J. L., & Padoy, N. (2025). Cholectrack20: A multi-perspective tracking dataset for surgical tools. CVPR25.
>
> ### W3. "The highlighting in Appendix Table 10 is inconsistent and paper cites 7 arXiv sources; are peer-reviewed publications for these papers available?"
>
> We have fixed the highlighting and replaced arXiv citations with available peer‑reviewed venues in the revision.
>
> ### Q1. "Claims no bounding boxes in CholecT50, but overlaps with CholecTrack20 providing bounding boxes; is annotation from both datasets comparable to presented annotations?"
>
> We appreciate the opportunity to clarify the relationship between these datasets. We confirm that the 5 test videos of CholecT50 indeed overlap with CholecTrack20. However, based on our download and inspection of CholecTrack20, the two datasets are not directly interoperable at the frame level. For example, for VID110, CholecT50 contains 2177 frames while CholecTrack20 contains only 1716 frames, and `000901.png` in CholecT50 is not the same scene as `000901.png` in CholecTrack20. Thus, the frame ID sequences are not in one‑to‑one correspondence; they simply originate from the same raw video. As a result, we cannot reliably align CholecT50 triplet labels with CholecTrack20 bounding boxes without an explicit mapping released by the original authors.
>
> Even if a clean alignment is provided in the future, the bounding boxes still exist only for the test set, so CholecT50 would remain unsuitable for fully supervised training of triplet detectors.
>
> ### Q2. "Claims inconsistent annotation boundaries in CholecT50, but relaxes temporal definition for momentary action to include up to 2 seconds before and after contact; how is consistent temporal annotation between annotators ensured? How is start and end defined when not exact time of contact?"
>
> We use a clear guideline and a multi‑round refinement strategy to ensure consistent temporal annotation. The “≤ 2 seconds before/after contact” rule is introduced to standardize annotators’ behavior across a very large dataset; without such a rule, boundaries would be more chaotic. Given the scale, 100% consistency is unrealistic, but Table 3 (Cohen’s κ = 0.82 with independent surgeons) shows that our protocol yields high agreement. In addition to the written guidelines, all annotations go through at least three rounds of checking and refinement (self‑check, cross‑check by another annotator, and expert review), which further improves the consistency of temporal boundaries. We also plan long‑term maintenance of ProstaTD to continuously fix residual issues.
>
> For your question about how start/end are defined when the current frame is not the exact contact time: annotators always first locate the true instrument–target contact frame and treat it as an anchor. If they notice that the current frame is not the exact contact, they scroll backward/forward to find that anchor frame, and then set the start and end boundaries relative to this anchor using the agreed “up to 2 seconds before/after” rule.
>
> ### Q3. "Merged triplets of aspirator sucking fluid, smoke and blood into second most common triplet; why merged, especially concerning intraoperative importance of bleeding?"
>
> We thank the Reviewer for highlighting the importance of smoke and blood. In real surgery it is often hard to clearly separate “aspirating blood” from “aspirating smoke.” For example, when scissors perform energy‑assisted cutting, bleeding and smoke usually appear together, and blood can be partially vaporized by the high‑temperature energy delivery, thereby contributing to the smoke plume. In such cases the aspirator is effectively removing blood and smoke at the same time, and at the frame level it becomes very unreliable to decide whether a given frame should be labeled as “mainly blood” or “mainly smoke.” We therefore merge these into a single category to avoid introducing noisy and irreproducible distinctions without adding robust clinical signal. This choice is also consistent with CholecT50, where blood and smoke are similarly merged.

---

> > ### Comment · Reviewer_wQHi · 2025-11-27
> >
> > I thank the authors for their detailed rebuttal and acknowledge the careful effort put into their response.
> > W1,2,3,Q1: sufficiently addressed
> >
> > Q2: I still do not understand exactly how the boundary is determined. There appear to be these magic 2-sec intervals that are applied somewhat arbitrarily. Could the authors provide a detailed protocol for this? Additionally, I remain unconvinced that the Kappa value truly improves as a result of the 2-sec rule. What is the minimal cohen's kappa possible if the annotations would match perfectly and only differ due to the 2-sec rule? would the kappa be higher without the rule?
> >
> > Q3: I agree with the difficulty between blood and fluid. But isn't the aspirator in the fluid vs in the air then suctioning one or the other? Vaporized blood could be counted as smoke.

---

> ### Author Response · Authors · 2025-11-20
> **Response to Reviewer wQHi (Part 2)**
>
> ### Q4. "Claims higher complexity than CholecT50 due to more triplets per frame, but CholecT50 has laparoscopic vs. ProstaTD robot-assisted surgeries where one tool not used, and over one quarter triplets are instrument with null action and null target; is complexity still higher when null triplets factored out?"
>
> Thank you for the question. Laparoscopic cholecystectomy is commonly used as a relatively basic / entry‑level procedure for early skills training of surgical residents [2,3]. In contrast, the Hospital Authority’s operation classification [4] places prostatectomy at a higher class than cholecystectomy, and in our setting robot‑assisted surgery is usually restricted to surgeons with higher professional rank and requires additional formal robotic training beyond standard laparoscopy. At the same time, our Table 3 also shows that laparoscopic cholecystectomy in CholecT50 can be completed with only three instruments, which further reflects the relative simplicity of the instrument setup compared with ProstaTD.
>
> Regarding the “null” triplets, these mostly occur because we explicitly define standardized start and end points for active actions. This naturally leaves intervals where an instrument is present but not performing a meaningful action (for example, pausing, waiting, or minor adjustment), and we label these as null. Surgeons do not operate continuously throughout the whole case, so such null periods are expected. If temporal boundaries are not clearly defined, many of these intervals can be implicitly absorbed into action segments instead of being labeled as null.
>
> Besides, based on our surgeons' inspection, the cholecystectomies in CholecT50 also appear to be relatively uncomplicated case selection (for example, with fewer arterial variation, adhesions, and inflammation), which may further reduce the complexity of the surgery. In contrast, ProstaTD includes a more challenging case mix (for example, some cases require meticulous lymph node dissection along the iliac vessels or involve markedly enlarged prostates), so overall the surgical scenarios in ProstaTD are more complex.
>
> [2] Koulas, S. G., Tsimoyiannis, J., Koutsourelakis, I., Zikos, N., Pappas-Gogos, G., Siakas, P., & Tsimoyiannis, E. C. (2006). Laparoscopic cholecystectomy performed by surgical trainees. JSLS. https://pmc.ncbi.nlm.nih.gov/articles/PMC3015736/
>
> [3] Sousa, J. H. B. D., Tustumi, F., Steinman, M., & Santos, O. F. P. D. (2021). Laparoscopic cholecystectomy performed by general surgery residents. Is it safe? How much does it cost?. Revista do Colégio Brasileiro de Cirurgiões. https://pubmed.ncbi.nlm.nih.gov/34008798/
>
> [4] Hospital Authority. Operations classification. https://www3.ha.org.hk/fnc/operations.aspx?lang=ENG, 2025. Online resource.
>
>
> ### Q5. "All these metrics only distinguish correct and incorrect triplets?"
> We agree that our triplet‑level metrics (mAP, Precision, Recall, F1) are binary in the sense that any error in instrument, verb, or target makes the whole triplet incorrect. To partially go beyond this 0/1 view at the triplet level, we additionally report component‑wise metrics for instruments, verbs, and targets (see Table 4 and Table 8), which measure mAP separately for each component. These results already show that the three components have different difficulty levels and error patterns, rather than being collapsed into a single triplet score.
>
> ### Q6. "Does impact of confusion differ between wrong tool vs. confusing action?"
> We sincerely thank the reviewer for raising this question. Our current binary triplet metrics do not encode differences in “importance” between components and treat all error types uniformly, primarily to remain comparable with prior work. Conceptually, however, not all components are equally critical. The instrument naturally tends to be the most important component, while the action and target are often of comparable importance. From a clinical and semantic perspective, correctly identifying the instrument is more critical than distinguishing fine-grained action labels or even the exact manipulated structure (target).
>
> ### Q7-1. "Could an ordinal metric like estimated cost provide insights into confusion of instruments, actions, or targets?"
>
> We thank the reviewer for this thoughtful suggestion. Our current analysis already offers insights into confusion patterns: component-wise mAP (Tables 4 and 8) shows weaker performance on actions and targets compared to instruments, and the triplet-level confusion matrix highlights that most errors stem from actions and targets, not instruments. For instance, thin, low-contrast objects like thread often lead to predictions such as (grasper, null, null) instead of (grasper, grasp, thread).

---

> ### Author Response · Authors · 2025-11-20
> **Response to Reviewer wQHi (Part 3)**
>
> ### Q7-2. "Could an ordinal metric like estimated cost provide insights into confusion of instruments, actions, or targets?"
>
> To further analyze error patterns, we now include separate single confusion matrix for instruments, actions, and targets (Appendix F.4, Figure 10). These reveal that instrument confusion is rare, while action pairs like Coagulate/Retract (42%) and Suck/Grasp (38%) are frequently confused, and non-organ targets (e.g., catheter, gauze) are often missed or misclassified.
>
> Besides, motivated by the reviewer’s question, we additionally explore an ordinal, cost‑aware metric that reflects the different importance of each component (bbox, instrument, action, and target), rather than treating all errors equally. In particular, we encode the ordinal preference “instrument > action ≈ target” directly in the scoring rule, and we follow the standard detection protocol to associate predictions with ground‑truth boxes: predictions are sorted by confidence and greedily matched to at most one ground‑truth box each (per image), with each ground truth used at most once, similar to mAP evaluation.
>
> We first perform greedy IoU‑based matching with threshold 0.5. For each prediction, if it cannot be matched to any ground‑truth box with IoU ≥ 0.5, its cost‑aware score is 0:
>
> $score_{IVT}$ = 0 if no matched GT with IoU ≥ 0.5, and $s_{ivt}$ otherwise.
>
> For matched pairs $s_{ivt}$, we assign partial credit to each component:
>
> $$
>     s_{ivt} = 0.1 +
>           0.5 \cdot \mathbf{1}[\text{instrument}] +
>           0.2 \cdot \mathbf{1}[\text{action}] +
>           0.2 \cdot \mathbf{1}[\text{target}].
> $$
> Here, the constant $0.1$ reflects that the bbox is already correct under the IoU $\ge 0.5$ condition, and thus indicates that at least “something” is happening in that region, even if the instrument, action, or target labels are not fully correct. The weight $0.5$ encodes the higher ordinal importance of the instrument, while actions and targets share a smaller but equal weight ($0.2$ each), and the weights sum to 1.0. Based on these per-prediction scores, we compute cost-aware “precision-like” and “recall-like” quantities by simple averaging over predictions and ground-truth instances, and then summarize them with a standard F1-style score (details in Appendi F.6):
> $$
> F1_{\text{cost}} = \frac{2 \cdot P_{\text{cost}} \cdot R_{\text{cost}}}{P_{\text{cost}} + R_{\text{cost}} + \varepsilon},
> $$
> In our analysis, these scores are further averaged over IVT classes, yielding a single cost‑aware F1 per component. This cost‑aware metric is used as a complementary analysis tool to standard mAP, providing an ordinal and decomposable view of how much error comes from instruments, actions, and targets. Below we show a preliminary cost‑aware analysis on our methods:
>
> Table: Cost‑aware F1 scores (5‑fold average over IVT classes) for bbox, instrument, action, target, and overall IVT.
> | Method        | $F1_{\text{cost}}^{\text{Bbox}}$ | $F1_{\text{cost}}^{\text{I}}$ | $F1_{\text{cost}}^{\text{V}}$ | $F1_{\text{cost}}^{\text{T}}$ | $F1_{\text{cost}}^{\text{IVT}}$ |
> |--------------|:---------------------------------:|:-----------------------------:|:-----------------------------:|:-----------------------------:|:-------------------------------:|
> | RT-DETR    | 0.0491                            | 0.2435                        | 0.0859                        | 0.0876                        | 0.4635                          |
> | YOLOv12      | 0.0503                            | 0.2508                        | 0.0856                        | 0.0836                        | 0.4676                          |
> | TDnet        | 0.0527                            | 0.2627                        | 0.0895                        | 0.0880                        | **0.4871**                         |
>
> From this table, we observe a qualitatively different behavior compared to the single-component mAP metrics. While single-component mAP is computed instrument-wise, action-wise, or target-wise, our new cost-aware metric is defined IVT-class-wise and thus penalizes all components jointly at the triplet level. As a result, even bounding boxes, instruments, actions, and targets all show noticeable room for improvement, especially for rare triplets where every component tends to perform poorly. Nevertheless, our proposed TDnet consistently achieves the best, indicating that its advantage remains under this stricter, triplet-aware evaluation.
>
> As for more complex ordinal schemes, for example prioritizing certain actions or targets more highly in specific clinical scenarios, this would require detailed input from expert surgeons and a task‑specific cost design. We view such class‑ or scenario‑specific ordinal cost modeling as promising future work beyond the scope of the current paper.

---

> ### Author Response · Authors · 2025-11-29
> **Further clarification for Reviewer wQHi**
>
> ### Q2. "Clarification of temporal boundaries and the 2-second tolerance rule.""
>
> Thank you for the opportunity to clarify this point. Our temporal boundaries are defined by a clinically informed and experience-driven protocol. The detailed annotation protocals are as follows:
>
> * For continuous actions (e.g., cutting the prostate): A triplet action begins only when the instrument is in contact with, or extremely close to, the target, and ends when it has been away from the target for more than 2 seconds. If this threshold is exceeded, the endpoint is backdated to the first frame where the instrument left the target. For example, consider a case where a grasper is retracting the prostate while scissors are continuously cutting. In the middle of this process, the scissors may stop for 5 seconds while waiting for the grasper to complete a new retraction before performing a deeper cut. With our 2‑second rule, this 5‑second stationary period is treated as a real pause, so the cutting action is split into two separate segments. Without this rule, the entire interval would be labeled as a single continuous “cut” action for the scissors, even though the instrument was inactive during the waiting period.
> This decision is based on clinical input and annotator experience: gaps shorter than 2 seconds are typically seen as minor adjustments, while longer gaps (≈3–5 seconds) are consistently judged as stop-and-restart events.
>
> * For momentary actions (e.g., cutting thread): We allow a 2-second window before and after the key contact frame to capture brief preparation and follow-through.
>
> Regarding Cohen’s kappa: since our labels are annotated and evaluated at 1 fps, annotators clearly understand that the 2‑second tolerance corresponds to a 2‑frame difference. Besides, without such a rule, a tool that is temporarily waiting while another instrument is completing an operation could still be counted as “actively performing the action” throughout, which would be misleading. Therefore, the rule serves as a clinically and experience‑driven guideline to standardize how annotators place temporal boundaries and to reduce spurious disagreement from boundary jitter. Without such a unified guideline, each annotator would follow their own subjective timing habits, and small boundary shifts would be counted as disagreements, lowering kappa even when there is no real semantic difference. More importantly, this would introduce inconsistent supervision for the model, making it harder for the network to learn a clear and clinically meaningful action.
>
> ### Q3. "Is the aspirator suctioning blood when in tissue, and smoke when in air?"
>
> We are pleased to further clarify this point. In routine practice, surgeons do not need to lift the aspirator tip into the air to remove smoke, as the suction is generally sufficient to affect the entire field of view. The aspirator is therefore typically kept close to, or in contact with, the tissue surface, where it often removes both blood and smoke simultaneously.
>
> During the annotation process, our expert annotators consistently reported that it was extremely difficult to reliably distinguish between “blood” and “smoke” in many frames, particularly when vaporized blood was present. To improve annotation reliability and inter-rater consistency, we therefore merged these two visually ambiguous phenomena into a single category in the current version of the dataset.

---

### Official Review · Reviewer_WshK · 2025-11-01

**Soundness:** 3
**Presentation:** 3
**Contribution:** 3
**Rating:** 4
**Confidence:** 4

**Summary:**

The paper introduces ProstaTD, a multi‑institutional dataset for fully supervised surgical triplet detection (⟨instrument, verb, target⟩) in robot‑assisted prostatectomy, with 71,775 frames, 196,490 instances, 89 triplet classes, 7 instruments / 10 actions / 10 targets, dense instrument bounding boxes, and clinically defined temporal boundaries. It also releases two tailored annotation tools and an evaluation toolkit, establishes a 5‑fold video‑level protocol, and provides a baseline (TDnet) with instance‑level self‑distillation. On the benchmark, TDnet achieves mAP_{IVT}@0.5 = 36.1% (vs. YOLOv12 34.3%) and F1 = 32.8% (Table 4–5). The work moves the community from frame‑level triplet classification (e.g., CholecT45/50) to full detection with explicit spatial/temporal supervision.

**Strengths:**

Data contribution: First full‑procedure, multi‑institutional, box‑supervised triplet dataset with standardized temporal boundaries; strong annotation process and κ=0.82.

Benchmark breadth: Comprehensive baselines (from SSD/Faster R‑CNN to RT‑DETR/YOLOv10‑12) with accuracy‑speed trade‑offs.

Tools & reproducibility: Open annotation apps + evaluation toolkit; 5‑fold protocol.

Empirical insight: Higher concurrency/scene density than CholecT50, stressing realistic IVT detection.

**Weaknesses:**

Domain generalization not tested: No leave‑one‑source‑out (e.g., train on PWH+ESAD, test on PSI‑AVA). Report would bolster the “multi‑institutional” claim.

Missing target boxes: Only instrument boxes are annotated; lack of target localization limits explicit interaction modeling; consider target boxes/segmentation or relation heads.

Method novelty modest: TDnet is effective but incremental; limited exploration of explicit relation modeling or stronger temporal modules.

**Questions:**

Can you report leave‑one‑source‑out IVT mAP/F1 to assess cross‑hospital generalization?

Any plan to add target boxes (or weak labels/segmentation) to better learn instrument–target spatial relations?

How robust are results under video‑wise metrics in the main text (now in Appendix F.2)?

---

> ### Author Response · Authors · 2025-11-20
> **Response to Reviewer WshK (Part 1)**
>
> ### W1 & Q1. "Domain generalization not tested: No leave-one-source-out (e.g., train on PWH+ESAD, test on PSI-AVA); report leave-one-source-out IVT mAP/F1 to assess cross-hospital generalization"
> As this work introduces a new dataset, our initial focus was on establishing strong intra-domain baselines to validate annotation quality and task difficulty. We originally deferred resource-intensive cross-domain evaluations, such as leave-one-source-out (LOSO) testing, to future work, given that preliminary analyses indicated minimal domain shifts across sources (PWH, ESAD, PSI-AVA), due to shared surgical protocols and similar imaging characteristics.
>
> Nonetheless, we agree that explicit LOSO evaluations provide a more rigorous assessment of cross-hospital generalization. In response, we have now included LOSO experiments below (and in the revision), training on two hospitals and testing on the third across all permutations.
>
> Table: Leave-one-source-out IVT performance across hospitals (train on two sources, test on the held-out source).
> | Leave-out Source （test set) | RT-DETR mAP_IVT 50 | RT-DETR mAP_IVT 95 | RT-DETR F1 | YOLOv12 mAP_IVT 50 | YOLOv12 mAP_IVT 95 | YOLOv12 F1 | TDnet mAP_IVT 50 | TDnet mAP_IVT 95 | TDnet F1 |
> |------------------|:-------------------:|:-------------------:|:---------:|:-------------------:|:-------------------:|:---------:|:-----------------:|:-----------------:|:--------:|
> | ESAD | 19.3 | 16.4 | 16.8 | 18.4 | 16.3 | 17.5 | 21.1 | 18.5 | 20.3 |
> | PSI-AVA | 22.1 | 20.1 | 19.9 | 20.6 | 18.7 | 20.7 | 23.1 | 20.6 | 23.4 |
> | PWH | 15.4 | 13.2 | 13.9 | 16 | 14.3 | 15.5 | 16.2 | 14.4 | 16.0 |
> | Average | 18.9 | 16.6 | 16.9 | 18.3 | 16.4 | 17.9 | 20.1 | 17.8 | 19.9 |
>
> These results show that all methods experience a substantial performance drop in the leave-one-source-out setting, indicating that cross-hospital IVT generalization is challenging on our dataset and underscoring its value as a benchmark for future domain generalization and continual learning methods.
>
> ### W2 & Q2. "Missing target boxes: Only instrument boxes annotated, lack of target localization limits explicit interaction modeling; any plan to add target boxes (or weak labels/segmentation) to better learn instrument–target spatial relations"
>
> We acknowledge that the current annotations focus on instrument bounding boxes, which limits explicit modeling of instrument-target spatial relations. This choice was deliberate due to the following challenges:
>
> 1. Many targets (e.g., prostate/bladder, liquid) lack clear, reproducible visual boundaries. Organs are often fuzzy and surgeon-dependent, while fluids/smoke have no definable edges, making reliable precise bounding boxes nearly impossible.
> 2. Establishing robust annotation guidelines for such fuzzy-boundary organs and ill-defined targets (e.g., blood) would require substantial effort and could constitute a standalone task, demanding extensive resources, potentially dozens of expert annotators over a year or more, though this is part of our future plans.
> 3. Instrument bounding boxes can support many practical purposes, as targets are typically in close proximity to instruments during interactions; additionally, the instrument serves as the initiator of the entire triplet, with its bounding box already including partial location information for actions and targets; thus, we can utilize instrument-proximal regions to extract and analyze relevant visual features for target recognition, effectively capturing spatial relations in the future work.
>
> Nevertheless, we view adding target boxes (or weak labels/segmentation) as a valuable future extension to enhance interaction modeling.
>
>
> ### W3. "Method novelty modest: TDnet effective but incremental, limited exploration of explicit relation modeling or stronger temporal modules."
>
> We would like to clarify that this is primarily a dataset and benchmark paper, with the main contribution being the ProstaTD dataset, following the structure of prior benchmark works such as CholecTrack20 [1] and SPORTS [2]. This setup ensures full reproducibility and provides a solid foundation for future method development.
>
> TDnet is introduced as a strong and insightful baseline, rather than a methodological contribution. As discussed in Appendix G.2, stronger temporal modeling is better leveraged with track identity labels in a detection+tracking pipeline, which also preserves real-time efficiency crucial for intraoperative use. Regarding explicit relation modeling, we fully agree on its importance and view it as a natural next step that ProstaTD enables. We plan to explore this direction in future work.
>
> [1] Nwoye, C. I., Elgohary, K., Srinivas, A., Zaid, F., Lavanchy, J. L., & Padoy, N. (2025). Cholectrack20: A multi-perspective tracking dataset for surgical tools. CVPR25.
>
> [2] Xia, H., Yang, Z., Zou, J., Tracy, R., Wang, Y., Lu, C., ... & Chen, H. SPORTU: A Comprehensive Sports Understanding Benchmark for Multimodal Large Language Models. ICLR25.

---

> ### Author Response · Authors · 2025-11-20
> **Response to Reviewer WshK (Part 2)**
>
> ### Q3. "how robust are results under video-wise metrics in the main text (now in Appendix F.2)?"
>
> We perform 5‑fold cross‑validation with splits at the video level (no video appears in both training and test). For each fold, we first compute the metric per video (by aggregating predictions over all frames of that video), then average across videos within that fold, so that each procedure contributes equally regardless of its duration. This video‑level definition is more robust than a frame‑level aggregation, which can be biased toward a few very long procedures or high‑frequency triplets.
>
> The final numbers reported are the mean and standard deviation across the 5 folds, which reflects the variability under different video‑wise splits. We also observe that the relative improvements of TDnet over the YOLOv12 baseline remain consistent under these video-wise metrics, indicating that our gains are not driven by a few specific videos. In addition, the dataset contains a reasonably diverse distribution of procedure lengths and triplet occurrences across videos, so no single video or small subset dominates the aggregated results. Taken together, the consistent performance across folds suggest that our findings under the video‑wise metrics are reasonably robust to the choice of split.
>
> **To Reviewer WshK: We thank Reviewer WshK for the thoughtful feedback. Many concerns appear to stem from a misinterpretation of our work as method-focused, while it is primarily a dataset and benchmark contribution. In the revision, we have addressed all points: added cross-hospital generalization results (W1 & Q1), clarified annotation choices and future plans (W2 & Q2), positioned TDnet as a baseline (W3), and detailed our robust video-wise evaluation (Q3). We respectfully request a re-evaluation in light of these clarifications and updates.**

---

### Official Review · Reviewer_TsFH · 2025-11-01

**Soundness:** 3
**Presentation:** 2
**Contribution:** 3
**Rating:** 6
**Confidence:** 4

**Summary:**

This paper presents ProstaTD, a surgical triplet dataset composed of 21 prostatectomy videos, annotated with instrument-action-target triplets. The dataset integrates multiple source datasets to increase diversity in instruments, actions, and surgical contexts. To support future research, the authors also propose TDnet, a baseline network with self-distillation and multi-task learning for triplet detection. Additionally, they provide two annotation tools and an evaluation toolkit. The dataset aims to support generalizable surgical action recognition and provide a benchmark for surgical triplet detection.

**Strengths:**

(1)ProstaTD integrates multiple sources (ESAD, PSI, PWH), covering different surgical styles, instrument usage, and rare triplets, increasing clinical coverage and the complexity of real-world surgical scenarios.
(2)The authors developed dedicated software (Triplet-LabelMe and SurgLabel) supporting structured triplets, temporal propagation, and batch operations. Both tools are open-source, significantly accelerating surgical video annotation and improving reproducibility.
(3)The ivtdmetrics package unifies I-V-T and IVT metric computation, supports video-wise evaluation, mAP@50:95, and multi-component statistics, facilitating fair comparisons across models.
(4)ProstaTD and its accompanying tools are planned for open-source release, making high-quality surgical triplet data and annotation tools widely accessible for algorithm development and benchmarking.

**Weaknesses:**

(1)A large portion of the manuscript is devoted to describing the dataset composition, annotation tools, and triplet statistics, while the experimental contribution is relatively limited.
(2)Although the TDnet baseline is proposed, the comparisons are primarily against standard detectors. The paper lacks exploration of different architectures, hyperparameter sensitivity, or deeper ablation studies. Additionally, it does not demonstrate the dataset’s utility on other downstream tasks such as surgical phase recognition or multi-instrument tracking.
(3)While the long-tail distribution is discussed, the paper lacks systematic strategies or metrics to evaluate model generalization on rare or unseen triplets. Additional ablation studies or case analyses would strengthen the argument regarding the dataset’s ability to support learning from rare events.
(4)Although the dataset is claimed to be procedure-agnostic, there are no experiments demonstrating its transferability to other types of surgeries. The paper lacks quantitative evidence or experiments to support the dataset’s claimed generalizability.

**Questions:**

The paper is relatively long, with overly detailed and occasionally repetitive descriptions of dataset statistics, annotation tools, and evaluation protocols. It is recommended to consolidate the key points and emphasize the core contributions, namely the dataset, the TDnet baseline, and the evaluation experiments, to improve readability.

---

> ### Author Response · Authors · 2025-11-20
> **Response to Reviewer TsFH (Part 1)**
>
> ### W1. "Large portion devoted to dataset composition, annotation tools, triplet statistics vs. limited experimental contribution"
> This work is a dedicated dataset contribution. Consequently, detailed exposition of composition, annotation protocols, and statistics is essential for reproducibility and future benchmarking. This structure mirrors established high-impact benchmarks like Cholectrack20 [1] and SPORTU [2].
> To address the concern regarding experimental scope, we have significantly expanded the validation in the revision (see the updated PDF). We now include comprehensive ablations, fusion-weight sensitivity analysis, and cross-hospital generalization experiments. These additions rigorously validate the dataset's utility without compromising the necessary depth of its characterization.
>
> [1] Nwoye, C. I., Elgohary, K., Srinivas, A., Zaid, F., Lavanchy, J. L., & Padoy, N. (2025). Cholectrack20: A multi-perspective tracking dataset for surgical tools. CVPR25.
>
> [2] Xia, H., Yang, Z., Zou, J., Tracy, R., Wang, Y., Lu, C., ... & Chen, H. SPORTU: A Comprehensive Sports Understanding Benchmark for Multimodal Large Language Models. ICLR25.
>
>
> ### W2. "TDnet baseline comparisons against standard detectors; lacks architectures, hyperparameter sensitivity, deeper ablations, downstream tasks like surgical phase recognition or multi-instrument tracking"
>
> As a foundational dataset contribution, our priority is the rigorous construction of the benchmark resource. However, to contextualize TDnet, we explicitly compare it against surgical-specific models (TAPIR and MCIT-IG) in Tables 4 and 8, rather than limiting the evaluation to standard detectors.
>
> To address the request for deeper analysis, we have added the following experiments below (also in the updated paper): 1) Architecture Generalization: We tested YOLOv10 and YOLOv11 extractors. The TDnet head yields consistent mAP_IVT improvements across backbones, proving the gains are architecture-agnostic. 2) Hyperparameter Sensitivity: We analyzed MTL self-distillation fusion weights. The results confirm that the proposed setting (0.8:0.2) achieves the optimal mAP_IVT and that the method remains robust to moderate weight variations.
>
> Regarding downstream tasks such as phase recognition and multi-instrument tracking, these require distinct annotations and architectural designs. They fall outside the scope of this dataset release and remain subjects for future work.
>
> Table: Ablations of the TDnet MTL head on different extractors under five-fold cross-validation.
> | Method                    | mAP_IVT 50       | mAP_IVT 95       |
> |---------------------------|------------------|------------------|
> | YOLOv10                   | $34.3_{4.1}$     | $31.8_{3.5}$     |
> | TDnet with YOLOv10 extractor | $35.1_{3.9}$ | $32.6_{3.3}$     |
> | YOLOv11                   | $34.1_{3.7}$     | $31.5_{3.3}$     |
> | TDnet with YOLOv11 extractor | $34.9_{3.5}$ | $32.3_{3.1}$     |
>
>
> Table: Sensitivity of TDnet to MTL self‑distillation fusion weights on ProstaTD fold1.
> | Fusion Weights    | mAP_IVT 50 | mAP_IVT 95 |
> |-------------------|:----------:|:----------:|
> | 0.0 : 1.0         |   40.9     |   37.8     |
> | 0.2 : 0.8         |   40.8     |   37.7     |
> | 0.5 : 0.5         |   41.3     |   38.0     |
> | 0.8 : 0.2 (TDnet) | **41.6**   | **38.2**   |
> | 1.0 : 0.0         |   41.4     |   38.1     |
>
>
> ### W3. "Long-tail distribution discussed but lacks strategies/metrics for rare/unseen triplets generalization; needs ablation studies/case analyses"
> As discussed in Appendix F.5 (Per-Class Average Precision of IVT Components), we have indeed demonstrated TDnet's effectiveness on both frequent and rare classes. To further support this, we conducted a separate mAP evaluation on rare triplets with fewer than 300 instances. Results (see table below) show that TDnet outperforms YOLOv12, suggesting improved generalization to long-tail IVT combinations. These new results have been added to Appendix F.3 of the revised manuscript. Regarding the case studies, Appendix F.4 already analyzes examples such as (grasper, grasp, thread), often misclassified due to the thread’s thin and low-contrast appearance.
>
> Table: Five-fold cross-validation performance on rare triplets (fewer than 300 instances in the dataset).
> | Method   | mAP_IVT@0.5 (Global) | mAP_IVT@0.5 (Video-wise) |
> |----------|:--------------------:|:------------------------:|
> | YOLOv12  | 19.8                | 20.7                    |
> | TDnet    | **20.6**            | **22.1**                |

---

> ### Author Response · Authors · 2025-11-20
> **Response to Reviewer TsFH (Part 2)**
>
> ### W4. "Procedure-agnostic claim but no transferability experiments to other surgeries; lacks quantitative evidence"
>
> Thank you for this insightful feedback. In the current paper, we only claim that our annotation software and evaluation toolkit are procedure‑agnostic (i.e., they can be directly applied to other surgeries), rather than claiming that the dataset itself already covers multiple procedures.
>
> In addition, TDnet is also procedure‑agnostic by design, as its architecture uses multi‑task learning and self‑distillation to model instrument–action–target (IVT) triplets without encoding any procedure‑specific prior or requiring surgery‑specific customization. While our dataset is scoped to prostate surgeries, it focuses on transferable elements, including ubiquitous instruments (e.g., graspers, scissors), common actions (e.g., grasp, cut), and targets that include both prostate‑specific structures (extendable to nearby organs such as the bladder) and generic entities (e.g., catheter, gauze, thread).
>
> ### Q1. "Paper long with detailed/repetitive dataset statistics, annotation tools, evaluation protocols; consolidate to emphasize dataset, TDnet, experiments"
>
> As suggested, we streamlined the manuscript by emphasizing core contributions, including enhancing dataset details and moving the TDnet key architecture to the main text, to improve clarity and readability in the revision.

---

### Author Response · Authors · 2025-11-20
**Overall response to reviewers**

Dear Reviewers,

We sincerely thank the chairs and reviewers for their thoughtful feedback and the time invested in reviewing our paper. Your comments were extremely helpful in improving both the clarity and quality of our work. In response, we have carefully revised the manuscript and added new analyses and experiments. Key updates include:

* Reorganized the paper by moving core TDnet architectural details from the appendix into the main text, while streamlining descriptive sections to improve clarity and focus. （Section 6)
* Added new ablations, including TDnet with YOLOv10/YOLOv11 extractors, a hyperparameter sensitivity study on MTL self‑distillation fusion weights and Rare-triplet mAP comparison. (Appendix F.3)
* Included single component normalized confusion matrices analysis on ProstaTD. （Appendix F.4 and Figure 10)
* Introduced a cost‑aware IVT metric assigning different weights to bounding box, instrument, action, and target errors for fine‑grained error analysis. (Appendix F.6)
* Performed leave‑one‑source‑out cross‑hospital experiments to evaluate domain generalization across PWH, ESAD, and PSI‑AVA. (Appendix F.7)
* Clarified and expanded the Limitations and Future Work section. (Appendix G.3 and G.4)
* Replaced arXiv citations with peer-reviewed conference/journal references where available.

Detailed responses are provided inline with each reviewer comment. We truly appreciate your insights and are happy to make further revisions if needed.


Sincerely,

The Authors

---

### Author Response · Authors · 2025-11-28

Dear Reviewers,

Just a gentle reminder to take a look at our rebuttal when convenient. If any concerns remain, we are very happy to clarify.

Thank you sincerely for your time and effort.

The Authors

---

### Meta-Review · Area_Chair_nWPV · 2026-01-10

**Summary:**

The reviewers provided many good comments for the manuscripts. One key point is the work’s model contribution. Though the authors claim the paper mainly as a dataset contribution, throughout the review process, the authors seemed to have demonstrated good results for their proposed model (TDNet).

The remaining questions about the dataset is the missing of target bounding box, the class imbalance issue, and how generalizable the current data collection method.

Otherwise, I believe the authors did a thorough job in answering reviewers questions, and most of the questions have been addressed well by the authors – so long as we contend that the paper’s main contribution is about creating a new dataset.

Hence I recommend an accept (poster) for this work.

**Reviewer Concerns:**

The reviewers raised many concerning points, such as
- The technical contribution of the work beyond dataset
- In what use context would having the additional bounding box inf be useful
- The contribution over existing work (CholecTrack20) that also includes bounding box
- The overclaim on their contribution on improving prior work’s inconsistency issues
- The overclaim of the complexity of the dataset
- Missing the target boxes, hence missing the opportunity to capture instrument-target spatial relationship
But the authors have provided reasonable answers to all of them.

The remaining questions about the dataset is the missing of target bounding box, the class imbalance issue, and how generalizable the current data collection method. Though the authors provided some answers, I don't think they are very convincing. But this should not diminish the dataset contribution of the work itself.

**Reviewer Scores:**

Most reviewers will probably keep their score as is.

---

### Decision · Program_Chairs · 2026-01-26

Accept (Poster)